# AutoSciDACT: Automated Scientific Discovery through Contrastive Embedding and Hypothesis Testing

**Samuel Bright-Thonney**[1,2]     **Christina Reissel**[1]     **Gaia Grosso**[1,2]     **Nathaniel Woodward**[1,3]
**Katya Govorkova**[1]     **Andrzej Novak**[1]     **Sang Eon Park**[1]     **Eric Moreno**[1]     **Philip Harris**[1,2]
[1]Department of Physics, Massachusetts Institute of Technology
[2] The NSF AI Institute for Artificial Intelligence and Fundamental Interactions
[3] Department of Physics, University of Wisconsin, Madison

## Abstract

Novelty detection in large scientific datasets faces two key challenges: the noisy and high-dimensional nature of experimental data, and the necessity of making *statistically robust* statements about any observed outliers. While there is a wealth of literature on anomaly detection via dimensionality reduction, most methods do not produce outputs compatible with quantifiable claims of scientific discovery. In this work we directly address these challenges, presenting the first step towards a unified pipeline for novelty detection adapted for the rigorous statistical demands of science. We introduce AutoSciDACT (Automated Scientific Discovery with Anomalous Contrastive Testing), a general-purpose pipeline for detecting novelty in scientific data. AutoSciDACT begins by creating expressive low-dimensional data representations using a contrastive pre-training, leveraging the abundance of high-quality simulated data in many scientific domains alongside expertise that can guide principled data augmentation strategies. These compact embeddings then enable an extremely sensitive machine learning-based two-sample test using the New Physics Learning Machine (NPLM) framework, which identifies and statistically quantifies deviations in observed data relative to a reference distribution (null hypothesis). We perform experiments across a range of astronomical, physical, biological, image, and synthetic datasets, demonstrating strong sensitivity to small injections of anomalous data across all domains.

## 1 Introduction

Scientific discovery is often characterized by serendipity: an unexpected observation turns out to have a profound impact on a field, leading to rapid progress or discovery. Today's data-rich scientific landscape is potentially brimming with curious or unexplained observations, but the scale and complexity of available data increasingly obscures genuine novelties behind statistical noise or incidental fluctuations. As scientific datasets continue to grow, so too grows the challenge of uncovering meaningful unexplained phenomena.

The scientific method traditionally provides a structured framework for discovery, encompassing observation, inquiry, research, hypothesis formulation, experimentation, and conclusion (top row of Fig. 1). Effective implementation of this method relies on human intuition and domain expertise to identify relevant observables and devise meaningful experiments. However, given the magnitude of modern datasets, the process would significantly benefit from automated tools to efficiently identify the most promising regions for discovery.

39th Conference on Neural Information Processing Systems (NeurIPS 2025).

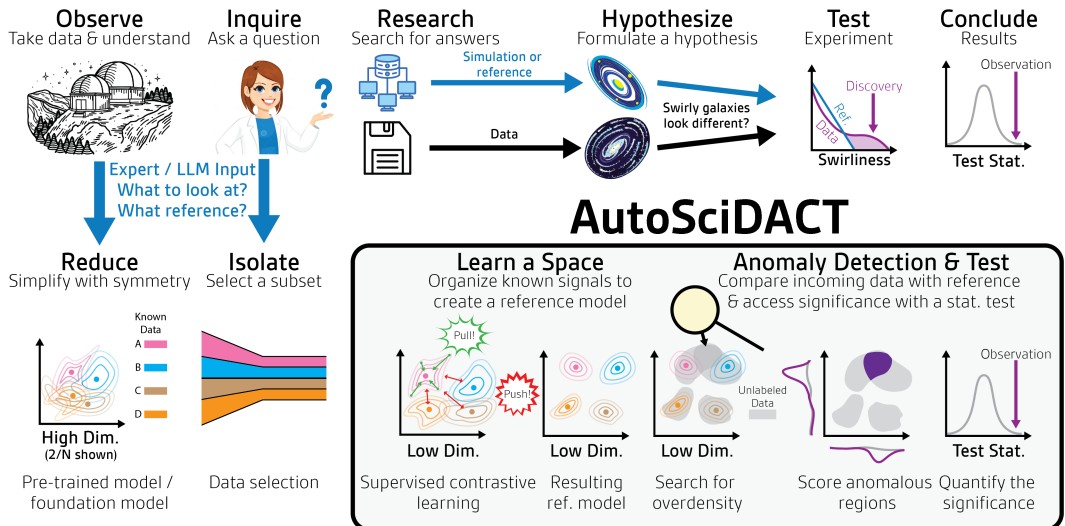

Figure 1: Illustration of the scientific method (top row) and the AutoSciDACT pipeline (bottom row), emphasizing the corresponding methodological steps implemented within AutoSciDACT.

To address this challenge, it is essential to develop methods that intelligently prioritize informative regions—areas where genuine scientific surprises are most likely to emerge. Traditional feature-engineering approaches are human-driven and domain-specific, limiting scalability and generalizability. Recent advances utilizing agentic AI systems and large language models can partially automate aspects of scientific inquiry [1–3], yet still lack integrated frameworks capable of rigorous, automated hypothesis testing and validation.

We introduce AutoSciDACT (Automated Scientific Discovery with Anomalous Contrastive Testing), a pipeline that parallels the scientific method and streamlines scientific inquiry by automating key steps of the scientific discovery process. AutoSciDACT streamlines the phases of data reduction, hypothesis formulation, and statistical testing (bottom row of Fig. 1) by deploying contrastive learning together with the New Physics Learning Machine (NPLM) [4]. Contrastive learning is used to reduce raw, high-dimensional datasets into expressive low-dimensional feature embeddings while NPLM provides a statistically robust mechanism for identifying and quantifying novel structures within this learned embedding space. A key insight with AutoSciDACT is the effective, automated incorporation of domain expertise as a tool to reduce the dimensionality to a small number of well-behaved features, making it possible to construct a robust statistical model. Using NPLM, our pipeline systematically compares incoming data with reference distributions of known (background) data, finding the most anomalous regions and quantifying their statistical significance.

We validate our approach using synthetic benchmarks and real-world datasets from astronomy, physics and biomedical domains. AutoSciDACT reliably detects meaningful novelties while remaining robust against spurious variations. Our results demonstrate the promise of combining structured contrastive learning methods with automated statistical hypothesis testing to accelerate scientific discoveries.

**Contributions**  Our main contributions can be summarized as follows:

- An end-to-end pipeline for novelty discovery in scientific datasets that is readily transferable across domains.

- A principled procedure for incorporating scientific simulations, hand-labeled data, and expert knowledge into a contrastive dimensionality reduction pipeline.

- A statistically rigorous framework for *quantifying the significance* of observed anomalies, beyond simply flagging anomalous datapoints.

- A realistic demonstration of novelty detection in four disparate scientific domains.

## 2    Related Work

**Contrastive Learning**  Contrastive learning is a powerful tool for learning expressive, low-dimensional data representations. Self-supervised methods such as SimCLR [5], MoCo [6], VI-CReg [7], and Barlow Twins [8] use data augmentations (e.g., blurring, cropping) to promote semantically meaningful and well-separated embeddings. Supervised contrastive learning [9] uses class labels to define positive pairs, more efficiently capturing semantic relationships between data-points but requiring labeled datasets to train. Its applications have expanded to structured data [10], multimodal inputs [11], and domain-specific tasks [12], demonstrating broad utility.

**Contrastive Anomaly Detection**  Several existing methods leverage contrastive embeddings to search for out-of-distribution (OOD) data, but primarily focus on identifying *individual* anomalous instances for e.g. industrial applications. In contrast, AutoSciDACT (our method) is tailored for scientific contexts and makes *statistical* statements about the presence of anomalous data, identifying distribution-level deviations relative to a baseline expectation (null hypothesis). Existing approaches include Refs. [13–21], all of which rely primarily on the AUROC for identifying OOD points as a figure of merit. They do not attempt to statistically quantify observations of OOD data, as is required in the scientific context. CADet [13] is nearest to our setup in using a two-sample test, but focuses again on AUROC. In scientific contexts, various combinations of contrastive learning and anomaly detection have been used for domain-specific applications – e.g. in astronomy [22, 23], histology, [24, 25] and particle physics [26, 27] – but, to our knowledge, no unified approaches have been proposed.

**Hypothesis Testing for Anomaly Detection**  Traditional goodness-of-fit (GOF) tests are powerful in univariate settings but struggle beyond that due to the curse of dimensionality. Simple multivariate extensions have limiting factors. The Mahalanobis distance-based test [28], for instance, assumes Gaussianity and is sensitive to the choice of covariance estimation, limiting its applicability in complex data regimes. Recent machine learning-based methods have introduced model-agnostic, data-driven alternatives to enable non-parametric, highly adaptable high-dimensional tests. To quantify distributional differences, Maximum Mean Discrepancy (MMD) [29–31] embeds distributions into a reproducing kernel Hilbert space, while the Classifier Two-Sample Test (C2ST) [32] uses a trained classifier's accuracy. Other methods include density-ratio estimation [33] and generative modeling frameworks that assess sample likelihoods [34]. In this work, we draw from statistical anomaly detection tests developed for high-energy physics, where sensitivity to subtle deviations is crucial. Autoencoders [35–46] and semi-supervised binary classifiers [47] have been widely used to score anomalies, but not statistically test them. The NPLM algorithm [48, 49] was introduced as an end-to-end score-and-test tool, outperforming classic GOF and classifier-based tests [50], showing sensitivity to a wide class of anomalies, and allowing incorporation of systematic (epistemic) uncertainties [51].

## 3    The AutoSciDACT Pipeline

The AutoSciDACT pipeline consists of two phases: pre-training and discovery. The aim of the pre-training phase is to learn an expressive, low-dimensional representation of a scientific dataset that retains key semantic features while reducing potentially hundreds or thousands of input dimensions to a handful. The discovery phase uses these embeddings in the NPLM anomaly detection and hypothesis testing framework to search for novelty in a scientific dataset. AutoSciDACT is designed for discovering *statistically significant* anomalies, prioritizing detection of distributional shifts (e.g. overdensities, distortions, outlier clusters) with respect to a background-only hypothesis, rather than instance-level anomalies. The power of NPLM (and any statistical test) degrades with data dimensionality, quickly requiring prohibitively large sample sizes to make statistically significant observations of small signals. As such, the reduction in the pre-training phase is critical. The bottom row of Fig. 1 summarizes the key steps and features of AutoSciDACT.

### 3.1    Pre-Training: Contrastive Embeddings

The backbone of our pipeline is an encoder $f_\theta : \mathcal{X} \to \mathbb{R}^d$ trained with contrastive learning to map raw data from its high-dimensional input space $\mathcal{X}$ to a low-dimensional representation in $\mathbb{R}^d$. Contrastive objectives are designed to maximize alignment between like inputs (positive pairs) while separating unlike inputs (negative pairs) in the learned space. We use the SimCLR framework [5], which trains

an encoder $f_\theta$ alongside a projection head $g_\phi$ (typically a small MLP) with the following contrastive loss:

$$\mathcal{L}_{\text{SimCLR}} = -\sum_{i \in \mathcal{B}} \log \frac{\exp(\text{sim}(\mathbf{z}_i, \tilde{\mathbf{z}}_i)/\tau)}{\sum_{j \neq i} \exp(\text{sim}(\mathbf{z}_i, \mathbf{z}_j)/\tau)}, \tag{1}$$

where $\mathbf{z} = g_\phi(f_\theta(\mathbf{x}))$, $(\mathbf{z}_i, \tilde{\mathbf{z}}_i)$ are a positive pair, $\text{sim}(\cdot, \cdot)$ is the cosine similarity, $\tau$ is a configurable temperature, and the sum in the denominator runs over all other pairings in a batch $\mathcal{B}$. Traditionally, positive pairs $(\mathbf{x}_i, \tilde{\mathbf{x}}_i)$ are constructed on-the-fly from inputs $\mathbf{x}_i \in \mathcal{B}$ using random augmentations that preserve semantic meaning. The projection head $g_\phi$ is discarded after training, with embeddings $\mathbf{h} = f_\theta(\mathbf{x})$ used for downstream tasks.

In AutoSciDACT we use *supervised* contrastive learning (SupCon) [9], which leverages labeled training data to create positive pairs from the same class and negative pairs from different classes. The training objective is a simple generalization of the SimCLR loss:

$$\mathcal{L}_{\text{SupCon}} = -\sum_{i \in \mathcal{B}} \frac{1}{|P(i)|} \sum_{p \in P(i)} \log \frac{\exp(\text{sim}(\mathbf{z}_i, \mathbf{z}_p)/\tau)}{\sum_{j \neq i} \exp(\text{sim}(\mathbf{z}_i, \mathbf{z}_j)/\tau)}, \tag{2}$$

where $P(i)$ is the set of all positive (same-class) pairs of input $i$ in the batch. Using labels to define $P(i)$ encodes a much richer notion of similarity from the full spectrum of a given input class, rather than having to indirectly learn (or fail to learn) important features from views of individual inputs. This also avoids the ill-defined question of identifying the "best" augmentations that promote expressive learned features. Practitioners in many scientific domains have ready access to large quantities of labeled training data from high-quality simulations or expert-labeled databases, so requiring labels is not often a significant bottleneck. When augmentations are desirable to encourage learning scientifically relevant meta-features (e.g. Lorentz invariance for particle physics datasets), or if class labels are unavailable, SupCon can easily incorporate augmented views in the positive set $P(i)$. In conjunction with labels, this offers a way to inject additional scientific domain knowledge via tailored augmentations.

In addition to the contrastive objective $\mathcal{L}_{\text{SupCon}}$ we include an optional supervised cross-entropy loss $\mathcal{L}_{\text{CE}}$, which we found beneficial for learning embeddings with a more regular structure and class separation. Our full loss function is thus

$$\mathcal{L} = \mathcal{L}_{\text{SupCon}} + \lambda_{\text{CE}} \mathcal{L}_{\text{CE}}, \tag{3}$$

where $\lambda_{\text{CE}} \sim 0.1$ - $0.5$ is set to make the classification objective sub-dominant.

## 3.2 Discovery: Anomaly Detection & Hypothesis Testing

In the discovery phase we use the embedding $f_\theta$ to process unseen datasets and search for anomalous clusters, overdensities, or outliers in the low-dimensional space. The search process is a classic scientific hypothesis test: a **reference** dataset $\mathcal{R}$ composed of known backgrounds is compared to an **observed** dataset $\mathcal{D}$ of unknown composition, and we seek to accept or reject the null hypothesis that $\mathcal{R}$ and $\mathcal{D}$ are identically distributed (i.e. there are no new phenomena in the observed data). We implement this test with NPLM, which in conjunction with the expressive learned embeddings enables extraordinary sensitivity to new signals.

**The NPLM algorithm** NPLM builds on the classical likelihood ratio test introduced by Neyman et al. [52], using a test statistic defined as:

$$t(\mathcal{D}) = 2 \max_{\boldsymbol{w}} \sum_{x \in \mathcal{D}} \log \frac{\mathcal{L}(x|\mathcal{H}_{\boldsymbol{w}})}{\mathcal{L}(x|\mathcal{H}_{\boldsymbol{0}})}. \tag{4}$$

A trainable model $f_{\boldsymbol{w}}(x)$ parametrizes a family of alternative hypotheses $\mathcal{H}_{\boldsymbol{w}}$ with respect to the null $\mathcal{H}_{\boldsymbol{0}}$ on inputs $x \in \mathbb{R}^d$, with a corresponding alternative density of the form:

$$p(x|\mathcal{H}_{\boldsymbol{w}}) = p(x|\mathcal{H}_0) \exp[f_{\boldsymbol{w}}(x)]. \tag{5}$$

This formulation enables a signal-agnostic approach: instead of specifying a particular signal model, the algorithm learns the deviation directly from data by solving a maximum likelihood problem reframed as a machine learning task. We follow the model introduced in [4], where the problem is

solved as a binary classification between the data of interest $\mathcal{D}$, labeled $y = 1$, and the reference sample $\mathcal{R}$, labeled $y = 0$. The model is a Nyström approximated kernel method

$$f_{\boldsymbol{w}} = \sum_{i=1}^{M} w_i k_i(x) \tag{6}$$

with $M \sim \sqrt{|\mathcal{D}| + |\mathcal{R}|}$ Gaussian kernels $k_i$ and trainable mixture coefficients $\{w_i \in \mathbb{R}\}_{i=1}^{M}$, and minimizing a regularized weighted binary cross-entropy

$$\mathcal{L}_{\mathrm{NPLM}}[f_{\boldsymbol{w}}] = \sum_{(x,y)} \left[ w_{\mathcal{R}}(1-y) \log\left(1 + e^{f_{\boldsymbol{w}}}\right) + y \log\left(1 + e^{-f_{\boldsymbol{w}}}\right) \right] + \lambda \sum_{i,j} w_i w_j k_i(x_j) \tag{7}$$

To ensure robustness, the size of the reference sample $|\mathcal{R}|$ is chosen to be substantially larger than the data $|\mathcal{D}|$ and is reweighted so that the expected yield under $\mathcal{H}_0$ matches the expected experimental one, which may differ from the observed size $|\mathcal{D}|$. This design choice makes the test sensitive to both shape and normalization deviations.[1]

Once the training is complete, the test statistic is estimated from the solution $f_{\hat{\boldsymbol{w}}}$ as

$$t_{\mathrm{NP}}(\mathcal{D}) = -2 \left( \sum_{(x,y)} w_{\mathcal{R}}(1-y) \left( e^{f_{\hat{\boldsymbol{w}}}(x)} - 1 \right) - y f_{\hat{\boldsymbol{w}}}(x) \right) \tag{8}$$

To calibrate the test, we estimate the distribution $p(t_{\mathrm{NP}}|\mathcal{H}_0)$ via pseudo-experiments ("toys"), generating (i.e. sampling from a larger pool) datasets under the null hypothesis and computing the corresponding test statistics to form an empirical distribution $\mathcal{T}_0$. The $p$-value is then evaluated empirically as:

$$p = \frac{1}{|\mathcal{T}_0|} \sum_{t \in \mathcal{T}_0} \mathbb{I}[t > t(\mathcal{D})]. \tag{9}$$

It can also be estimated asymptotically from the distribution of $\mathcal{T}_0$, which can be fit to a suitable $\chi^2$ distribution [4]. The asymptotic estimate is useful in cases where the deviation of $\mathcal{D}$ with respect to $\mathcal{R}$ is large (e.g. $Z = 5\sigma$), in which case the number of toys required for the empirical estimate would be prohibitively large.

The power of the NPLM test strongly depends on the choice of the kernels' width, as it determines the scale of distortions the model is sensitive to. To mitigate this feature and make the model more robust, we adopt an extended version of the algorithm introduced in [53], where multiple widths are considered and combined to obtain a final $p$-value. The authors of [53] explore several options for combining the tests based on "local" $p$-values, but in this work we choose the average of $p$-values as a rule. The average score is typically less powerful than the single "optimal" kernel, which can be considered a kind of "look-elsewhere" effect accounting for various kernel hypotheses.

We consider six different kernel widths for our experiments, with their precise numerical values chosen according to the distribution of pairwise distances between data points in the embedding space. More precisely, the first five values are the 1st, 25th, 50th, 75th, and 99th percentiles of the empirical pairwise distance distribution (computed with a subset data points from the training set); the last value is twice the 99th percentile, and it ensures sensitivity to out-of-distribution anomalies. This choice means the numerical kernel widths vary among datasets, so we denote them by their corresponding quantiles: $\sigma_{\mathrm{ker}} \in \{q_1, q_{25}, q_{50}, q_{75}, q_{99}, 2q_{99}\}$.

The NPLM procedure provides a flexible, multivariate, unbinned likelihood-ratio test that is agnostic to the source of the anomaly, making it well-suited for unsupervised anomaly detection tasks. Comparisons with alternative GoF approaches presented in [50] show the impressive sensitivity of the method to subtle distortions of the data density distribution. As for any GoF approach relying on density estimation from empirical samples, limitations arise when scaling the data dimensionality. In this work, we target the curse of dimensionality by compressing high-dimensional raw data with contrastive embeddings.

---

[1]This sensitivity is important in contexts where data collection windows determine $|\mathcal{D}|$, and deviations in event rates may signal anomalies.

## 4 Datasets

We demonstrate AutoSciDACT on a diverse collection of five synthetic, image, and scientific datasets. Each dataset contains a large collection of data from "background" (i.e. well-understood) classes that are used in the contrastive pre-training phase, along with a set of anomalous "signal" data. In the discovery phase we construct datasets $\mathcal{D}$ with small signal injections and use NPLM to detect the novel component. We briefly describe each dataset below, with additional details available in App. A. Due to space constraints, we defer two further studies to the appendix: a genomics task identifying novel butterfly hybrids from wing images (App. A.6), and searching for four-lepton decays of the Higgs boson in real LHC data (App. E).

**Synthetic Data** The synthetic dataset is designed to illustrate the core functionality of AutoSciDACT independent of details specific to scientific datasets. It consists of points $\mathcal{X} \subset \mathbb{R}^{D+M}$ with $D$ meaningful dimensions and $M$ noisy dimensions. The noisy dimensions are sampled from $\mathcal{U}(0,1)$, and the meaningful dimensions are populated by $N$ Gaussian clusters $\mathcal{N}(\boldsymbol{\mu}_i, \boldsymbol{\Sigma}_i)$ $(i = 1, \ldots, N)$ with means $\boldsymbol{\mu}_i \sim \mathcal{U}(0,1)$ and covariances $\boldsymbol{\Sigma}_i \sim \mathcal{U}(0, 0.5)$. The pairwise distances of the Gaussian clusters are adjusted such that a 1% injection of one cluster on top of 10k samples of another yields a deviation of 3.5 standard deviations. The full $D + M$-dimensional space is then randomly rotated to obscure the discriminating variables. The contrastive embedding is trained on $N - 1$ clusters with one held out as a signal, with the backbone architecture $f_\theta$ being a simple MLP.

**Astronomy** For an astronomical baseline we choose gravitational wave data recorded by the Laser Interferometer Gravitational-Wave Observatories (LIGO) in Hanford, WA and Livingston, LA [54]. Although gravitational waves from compact binary systems have been detected [55], many hypothetical sources remain unobserved, making the challenge particularly intriguing for anomaly detection methods. The data consist of 50 ms time-series signals from two channels - one for each interferometer - sampled at 4096 Hz (200 measurements per channel) [56, 57]. The different classes of data consist of pure background ($\sim$ gaussian noise), "glitches" (periods of short-duration transient instrumental noise), and six observed or hypothetical sources of astrophysical signals. A seventh signal class with "white noise burst" (WNB) waveforms is held out from pre-training and is injected as an anomaly in the discovery phase. The encoder architecture is a one-dimensional ResNet, following the technical setup explored in [58] for identification of binary black holes.

**Particle Physics** Our particle physics baseline is JETCLASS [59, 60], a large dataset consisting of simulated *jets*: energetic, collimated streams of $\mathcal{O}(100)$ particles that are produced in proton-proton collisions at the Large Hadron Collider (LHC). We use a subset of JETCLASS consisting of jets from quantum chromodynamics (QCD) processes (quark/gluon), top quark decays ($t \to bqq'$), and W/Z vector boson decays ($V \to qq'$). We hold out signal jets from boosted Higgs boson decays to bottom quarks ($H \to b\bar{b}$), inspired by recent measurements of $H \to b\bar{b}$ in the combined gluon fusion and vector boson production modes by the CMS experiment [61]. We use the Particle Transformer (ParT) architecture [60] – a variant of the Transformer architecture [62] adapted for particle physics – as the contrastive encoder.

**Histology** As an example from life sciences, we aim to identify abnormal tissue in histopathological images. The abundance of healthy tissue data and the difficulty in collecting samples with various abnormalities render histology particularly well-suited to anomaly-detection tasks. We use publicly available optical microscope images from stained tissue samples [63]. Our reference sample contains seven classes of tissue from mice (brain, heart, kidney, liver, lung, pancreas, spleen) and one class of normal liver tissue from rats. We aim to detect anomalous mouse liver tissue caused by non-alcoholic fatty liver disease (NAFLD). Inputs are 256x256 pixel (0.44μm/pixel) resolution tissue tiles extracted from the whole slide image with Masson's trichome staining. As a backbone, we train the best performing architecture from Ref. [25], EfficientNet-B0 [64].

**Images** We use the CIFAR-10 dataset [65], arbitrarily holding out class 1 as the anomaly and pre-training on the other nine classes. In the discovery phase of the pipeline, we use images from CIFAR-5m [66] to supplement the CIFAR-10 test set and expand the number of data points available for hypothesis testing.[2] We use a ResNet-50 encoder backbone with pre-trained weights [67], swapping out only the final fully connected layer with a slightly larger MLP and fine-tuning it on the CIFAR contrastive embedding task.

---

[2]CIFAR-5m was introduced in [66] and consists of images generated by a diffusion model trained on CIFAR-10, which were found to be nearly indistinguishable from CIFAR-10 by a pre-trained classifier.

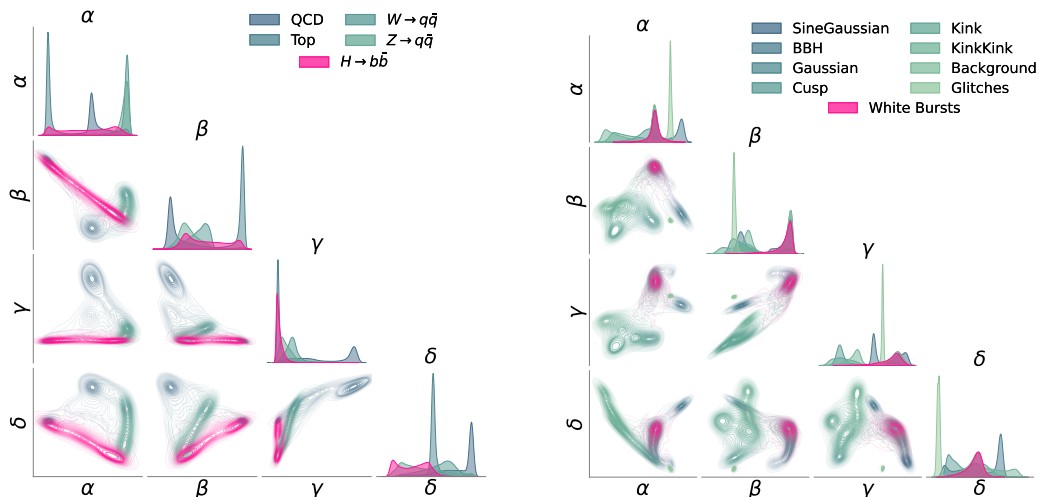

Figure 2: Contrastive embeddings for *Particle physics* (left) and *Astronomy* (right) datasets. The high-dimensional input is projected down to four dimensions ($\alpha, \beta, \gamma, \delta$). Background classes are shown in hues of blue and green, while the anomaly is overlaid in hot pink.

## 5 Experiments

**Embedding** We train and evaluate the AutoSciDACT pipeline on each dataset in the same manner, making only small adjustments in pre-training to adapt to the specifics of each dataset (see App. A for full details). We fix the embedding dimension to $d = 4$ for all encoders to put each on equal footing for NPLM, whose performance varies with input dimensionality. The choice of a low embedding dimension is made to ensure that statistical tests remain tractable, and to demonstrate that it is possible to obtain strong anomaly detection performance with a very compact representation (see App. B.3 for a study of larger embedding dimensions). In Fig. 2 we visualize the learned contrastive embeddings for JetClass and LIGO, with embeddings of the anomalous class - which was not included in the training - indicated in pink. The anomalous cluster in JetClass manifests as an extended and distinct cluster, while in LIGO it is an overdensity near a background-populated region. Flagging the latter would be challenging for traditional per-datapoint anomaly detection methods, but we will demonstrate that NPLM detects it as an overdensity.

**Anomaly detection & hypothesis testing** We follow a standardized procedure for signal injection, anomaly detection, and hypothesis testing for each dataset. As described in Sec. 3.2, we compile a reference sample $\mathcal{R}$ from the test set composed entirely of the known classes used in training. We then construct a "observed data" set $\mathcal{D}$, also from the known classes and in the same relative proportions as $\mathcal{R}$. We mimic the presence of novelty in $\mathcal{D}$ by injecting some number $N_S = f_S|\mathcal{D}|$ of anomalous signal datapoints from the held out class, where typically $f_S \lesssim 0.1$. For each injection rate $f_S$, we run 500 NPLM pseudo-experiments to populate a distribution of test statistics $t(\mathcal{D}; f_S)$, re-sampling $\mathcal{D}$ and signal injections each time.[3] To calibrate the test, we run 500 additional pseudo-experiments with $f_S = 0$ to populate a reference distribution of $t(\mathcal{D}|f_S = 0)$. The empirical and asymptotic $p$-value and $Z$-score are computed. For each dataset, we scan $f_S$ across a range of injection fractions and plot the resulting $Z$-scores in Fig. 3. The size and composition of $\mathcal{R}$ and $\mathcal{D}$ are fixed by the practical limitations of each dataset (i.e. the test set size), and $f_S$ is varied in a range where NPLM starts to become sensitive to the injected signal. This information is summarized in Table 1 in App. A. Each panel of Fig. 3 also includes results from three baseline statistical tests to compare with NPLM: two supervised tests that incorporate explicit knowledge of the signal, and a test based on the Mahalanobis distance [28].

**Supervised baselines** We use two fully-supervised baselines as an estimate of best-case anomaly detection performance. We denote them "supervised" and "ideal supervised", distinguishing the extent to which knowledge of the true signal is utilized. For the "supervised" baseline we train an MLP to identify the desired signal in the contrastive embedding space, while for "ideal supervised"

---

[3]In cases where the test datasets are large enough, we re-sample $\mathcal{R}$ as well. Full details are in Appendix A.

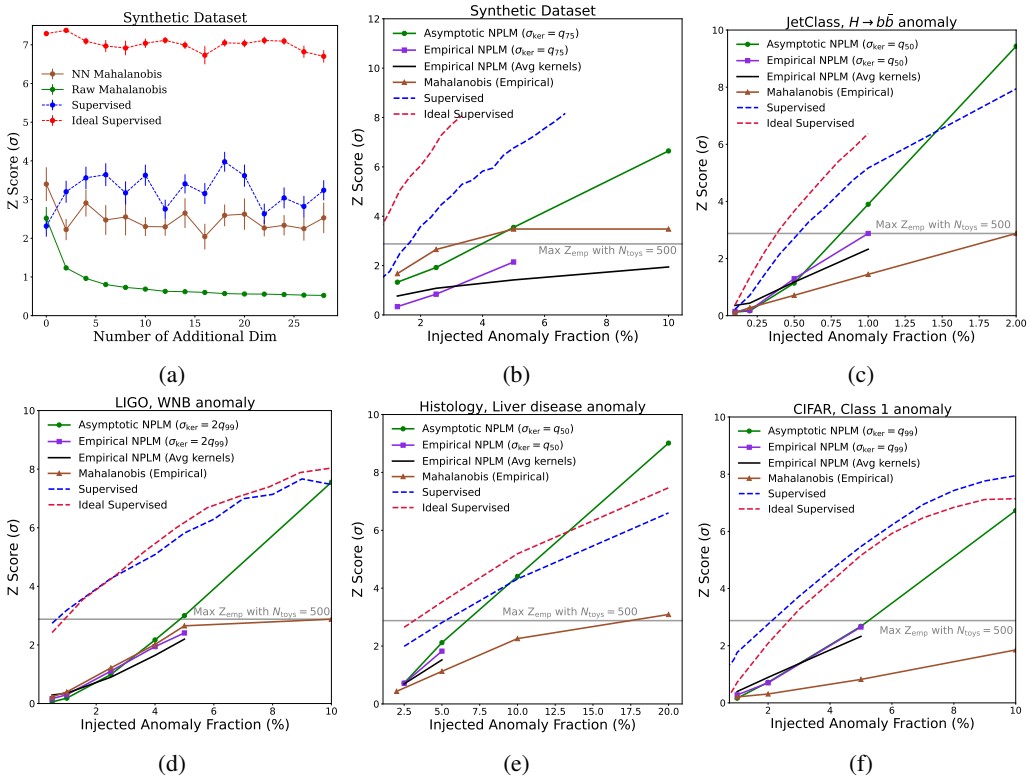

Figure 3: Statistical significances ($Z$ scores) of NPLM and other baseline methods for detecting various fractions of anomalous signals injected into background-dominated samples in (a) scanning additional random variables to the same Synthetic toy at a fixed fraction of 0.6% with 10k Background, and then for the fraction for (b) Synthetic (2k), (c) particle physics, (d) astronomy, (e) histology, and (f) image datasets. In all cases, NPLM is able to discover very small signals with high confidence. The upper limit of the empirical $Z$ scores is indicated by a gray line at roughly $2.88\sigma$ and is set by the fixed number of pseudo-experiments (500), so empirical numbers are not quoted beyond this point. The Asymptotic NPLM approximates the large pseudo-experiment limit at large $Z$ scores.

baseline we do the same but first *re-train* the contrastive embedding with the true signal added to the training set (the encoder has explicit knowledge of the signal). The reference $\mathcal{R}$ and observed $\mathcal{D}$ datasets are constructed the same way as NPLM, but the hypothesis test relies on more typical statistical methodology. We construct one-dimensional distributions of classifier scores $s$ (normalized to the range [0,1]), and use the points in $\mathcal{R}$ to construct a background-only shape template $f_R(s)$ and signal points to construct a signal template $f_S(s)$. We then perform a binned maximum likelihood fit to the classifier scores in $\mathcal{D}$ under the null ($H_0 : \mathcal{D} \sim a_1 p_{\mathcal{R}}(s)$) and alternative ($H_1 : \mathcal{D} \sim a_1 p_{\mathcal{R}}(s) + a_2 p_S(s)$) hypotheses and compute the test-statistic $\Delta\chi^2 = \chi^2_{H_0} - \chi^2_{H_1}$. We compute empirical and asymptotic[4] $p$-values and $Z$-scores over many pseudo-experiments [68].

**Mahalanobis baseline** As a comparison with analytic anomaly score, we use the Mahalanobis distance metric [28]. For each pseudo-experiment we compute the mean $\boldsymbol{\mu}_i$ and covariance $\boldsymbol{\Sigma}_i$ for the embeddings of each background class $i$ in $\mathcal{R}$. The Mahalanobis distance is then $d_{\text{Maha}}(\mathbf{x}, \mathcal{R}) = \min_i(\mathbf{x} - \boldsymbol{\mu}_i)^T \boldsymbol{\Sigma}_i^{-1}(\mathbf{x} - \boldsymbol{\mu}_i)$, and we define the test statistic as $t_{\text{Maha}}(\mathcal{D}) = \sum_{\mathbf{x} \in \mathcal{D}} d_{\text{Maha}}(\mathbf{x}, \mathcal{D})$. Empirical $p$-values and $Z$-scores are computed as before.

---

[4]$\Delta\chi^2$ is asymptotically $\chi^2$ distributed with one degree of freedom.

# 6 Discussion

## 6.1 Results

The results in Fig. 3 clearly demonstrate the power of the AutoSciDACT pipeline, with NPLM flagging highly statistically significant deviations ($Z \gtrsim 3$ or $p \lesssim 10^{-3}$) with signal fractions as low as 1%. The two supervised baselines provide a reasonable upper limit on the sensitivity to the signal given full knowledge of its distribution in the embedded space, and in some cases, NPLM performs near this limit. Beyond roughly $5\sigma$, some trends break down, but at this level of significance ($p \sim 10^{-7}$), discovery is extremely clear.

In all but the synthetic datasets, NPLM significantly outperforms the Mahalanobis baseline due to the flexible range of distortions and overdensities it is capable of modeling in the input space. The Mahalanobis test is best suited to cases where each background cluster is roughly normally distributed, and by construction is not sensitive to overdensities near the bulk of any given cluster. Since the synthetic dataset is constructed from Gaussian clusters, Mahalanobis is quite effective in this case. In Fig. 3(a), we leverage the computational efficiency of Mahalanobis distance by running over 100 toy synthetic datasets per point, comparing performance on raw versus embedded inputs. As is clear with the raw performance, additional random variables quickly destroy the sensitivity to hidden signals. Sensitivity is preserved across all numbers of random variables using the fixed-dimensional embeddings as inputs, as the large number of noisy dimensions has little impact on the quality of the embedding.

For both the LIGO and JetClass dataset, we approach the supervised limit at a $Z$-score of 3, which rivals or exceeds all anomaly detection algorithms within their respective domains [69–74]. While astronomy and particle physics have long leveraged statistically rigorous anomaly-detection techniques, their application to histology illustrates a successful transfer of methods across scientific disciplines. The results on the histology datasets align with the findings reported in [25], which demonstrate that embedding spaces constructed with label information outperform those based solely on data augmentations. With AutoSciDACT, we introduce a new method capable of detecting localized abnormalities that may be present in only a small fraction of tissue, a capability that is essential both for early detection of disease and for guiding pathologists' judgments on toxic compounds.

## 6.2 Limitations

**Domain knowledge**   Since AutoSciDACT relies exclusively on domain knowledge in the label information, its performance is highly correlated with the label quality. Although labeling is easy and accurate in some domains (e.g. simulations, or organ labels for histological patches), labeling large training subsets can be laborious or impossible. For all baseline results, we also assume equal distributions from all background classes in the reference distribution for both pre-training and discovery. However, the actual composition of the reference sample during discovery needs to resemble the one in the observed data, and may require additional input from domain experts. These are problems routinely solved by scientists, so they do not pose a major obstacle to implementing the pipeline.

**Embedding dimensionality**   Embedding into a small space ($d = 4$) limits expressivity, though the features learned in contrastive pre-training will typically be more useful than handpicked variables. This is most evident in the LIGO and CIFAR-10 results in Fig. 3, where the "ideal supervised" benchmark falls short of the supervised one when it should in principle do better. This is due to the density of a large number of classes, which struggle to be perfectly separated in the four-dimensional space. The "ideal" scenario, including an additional class in the learned space, exacerbates this problem. The embedding dimension can be reasonably scaled up (see App. B.3), but beyond a certain point (e.g. hundreds or thousands of dimensions) NPLM's sensitivity will degrade substantially due to the sparsity of the data.

**Domain shift and uncertainties**   In all experiments, we assume that the reference dataset correctly resembles the background distribution of the data. While this is an exact assumption in cases where it is possible to label subsets of data, the reference sample might contain domain shifts if it is constructed from data recorded under different conditions or from simulation. The impact of domain shift on contrastive embeddings has been studied in [75], and the inclusion of epistemic uncertainties within

both NPLM and the embeddings is possible [50, 76]. Extensions of AutoSciDACT, including domain shifts and estimation of the associated epistemic uncertainties, are left for future work.

## 6.3 Conclusion & Future Work

In summary, we have presented what is, to our knowledge, the first end-to-end scientific pipeline for novelty discovery in arbitrary datasets with a rigorous statistical foundation based on hypothesis testing. We show that using AutoSciDACT, we discover anomalous signals with high statistical significance ($\geq 3\sigma$) even when the data contains only a percent-level signal fraction and the dimensionality of the raw data is large. By applying AutoSciDACT to five different datasets from four different scientific domains, we prove the methods' universality and transferability, enabled by the strict decorrelation of expert knowledge encapsulated in label information from the actual analysis pipeline. For a comprehensive scientific outcome, incorporating potential domain shifts along with their associated uncertainties is essential. We plan to further extend AutoSciDACT through known extensions of our methods. More generally, by abstracting the scientific method, our approach presents a framework that automates scientific discovery, leading to the possibility of rapid, comprehensive, and rigorous scientific analysis on all data.

## 7 Acknowledgments

This work is supported by the National Science Foundation under Cooperative Agreement PHY-2019786 (The NSF AI Institute for Artificial Intelligence and Fundamental Interactions, http://iaifi.org/). CR acknowledges support under a Postdoc.Mobility fellowship from the Swiss National Science Foundation (SNF) (Grant No. 222340). Computations in this paper were run on the FASRC Cannon cluster supported by the FAS Division of Science Research Computing Group at Harvard University.

Following the publication in NeurIPS. It was pointed out that reference [12] was linked to an incorrect, AI-hallucinated article. We acknowledge this mistake, which resulted from an error in preparing the BibTeX citation using LLMs, prompting the LLM to generate the `BibTeX` citation with the correct first author and the first word of the title. The updated draft now includes the intended reference.

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

| Dataset | $|\mathcal{R}|$ | $|\mathcal{D}|$ | $f_S$ range (%) |
|---|---|---|---|
| Synthetic | 10,000 | 2,000 | 0.5 - 10 |
| Astronomy | 20,000 | 4,000 | 0.5 - 10 |
| Particle Physics | 50,000 | 10,000 | 0.1 - 2 |
| Histology | 6,296 | 500 | 2.5 - 20 |
| Images (CIFAR-10) | 9,000 | 1,800 | 1 - 10 |
| Genomics | 1,764 | 100 | 5 - 20 |

Table 1: The reference dataset size, observed dataset size, and range of injected anomaly fractions (relative to the background component in $\mathcal{D}$) for each dataset considered in our study. These parameters are used when running NPLM pseudo-experiments across a range of signal fractions to produce the results shown in Fig. 3.

## A  Dataset, Training, and Evaluation Details

All of the experiments presented in this paper were run on an academic computing cluster. The contrastive trainings were run on a single NVIDIA A100 GPU in all cases, and none took more than a few hours to compute. The kernel-based NPLM tests used the GPU-accelerated Falkon package [77, 78] and also ran on a single GPU, with a typical set of 100 toys with $|\mathcal{R}| = 10,000$ and $|\mathcal{D}| = 2,000$ taking 10-20 minutes. All other metrics such as the Mahalanobis test were computed on CPU nodes and were not a significant computational bottleneck.

Table 1 summarizes the reference dataset sizes $|\mathcal{R}|$, observed dataset sizes $|\mathcal{D}|$ and signal fractions $f_S$ used in the experiments presented in Sec. 5. These numbers are typically limited by the test set sizes for each dataset, and by requirement that $|\mathcal{R}|$ be significantly larger than $|\mathcal{D}|$ for NPLM. We fix the ratio at $|\mathcal{R}| = 5|\mathcal{D}|$, except for histology and genomics where the datasets are very small.

As mentioned in Sec. 3.2, the kernel size $\sigma$ is a configurable hyper-parameter of NPLM, and the performance varies somewhat as the kernel width changes. In practice, all NPLM pseudo-experiments are run with six different variations of the kernel width $\sigma = [0.1, 1.5, 2.6, 3.6, 4.9, 9.8]$. The four dimensional input data are standardized according to the mean and standard deviation of the reference sample $\mathcal{R}$, so these widths refer to a common scale. The smallest-width kernels are best at adapting to small, local features and distortions in the data, while the widest ones can capture excesses or outlier far in the tails away from the bulk background distribution. In Fig. 3 of the main text we present the asymptotic and empirical NPLM $Z$ scores corresponding to the best-performing kernel width, but we also show average empirical $Z$ of all six kernels in black. We plot full results for all kernel widths, both asymptotic and empirical $Z$ scores, in App. B.

### A.1  Synthetic Data

The synthetic dataset aims to broadly look at challenging datasets that are largely overlapping and high-dimensional. As part of that, we insisted on a series of core elements to ensure a robust construction. Namely, the chosen mixture of Gaussians

- signals are fully reproducible,
- the minimum pairwise optimized significance between clusters was 3.5 standard deviations; this was computed through the computation of a cumulative distribution about the mean of the Gaussian, assuming a scenario of 1 percent background in a sample size of 10000 events.
- All discriminating variables were randomly mixed among non-discriminating variables
- Gaussian means and sigmas are bounded in ranges of $[0, 1]$ and $[0.02, 0.5]$, respectively.

For each dataset, we generated 10k events in each of the data classes. Datasets ranging from 3 separate classes to 20 were generated, along with additional random variables ranging from 0 to 30 variables. Additionally, to address the variation of models, we generated roughly 100 separate random models for each point. The total number of models utilized is 3870.

With each model, two trainings are performed, using a simple 4-layer MLP with a separate MLP classifier with no output activation and a projector to compute the contrastive loss; there are a total

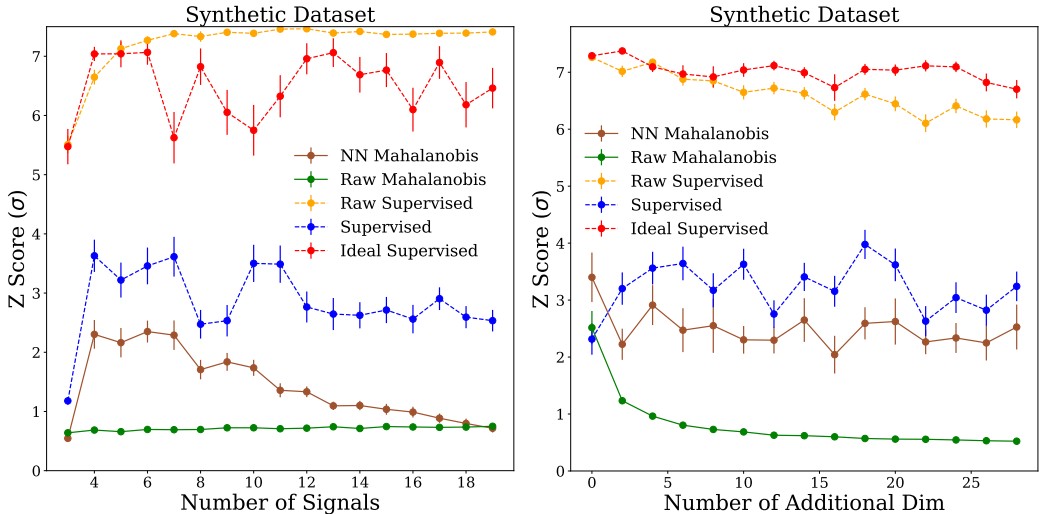

Figure 4: Z-score detection statistic for an anomaly using Mahalanobis distance, and supervised learning on the raw and learned embedded space. (a) We observe the impact of additional signals on sensitivity to find an anomaly. (b) We observe the change in sensitivity vs the number of additional random variables obtained from the average of 100 toys, adding an additional signal model and discriminating variables with each increment on the x-axis. The green corresponds to the Mahalanobis distance on raw inputs, the orange corresponds to a fully supervised algorithm on raw inputs, and the red corresponds to a supervised algorithm trained on the ideal contrastive space, where the signal is known. The blue shows the result of a supervised training (knowing signal) on a contrastive space where the signal is unknown; the brown shows the result of the Mahalanobis distance on the same space.

of 12k trainable parameters. The loss function used was SimCLR with $\tau = 0.01$ and $\lambda_{\text{CE}} = 0.5$, a learning rate of 0.001 with a batch size of 1000 is used along with a cosine annealing. Trainings are performed over 50 epochs and take roughly 5 minutes on a CPU. Similarly, for the supervised algorithms, a 4-layer MLP of roughly the same size was utilized.

Figure 4(a) presents the result of scanning over the toy models, computing the asymptotic Mahalanobis distance for the embedded space, the raw space, and applying a supervised algorithm on both, and with an additional supervised algorithm on the embedded space trained with the hidden signal. The left plot adds an additional signal and an additional discriminating dimension with each variable. Here, we observe that at least four signals are needed to span the space, and high sensitivity is observed, which gradually goes down as the confusion and density from so many signals make it hard for a specific point to separate itself. The right plot shows the impact of additional random dimensions on the data. We observed that either embedding or a supervised algorithm is sufficient to overcome a loss of sensitivity present from just adding random, fluctuating dimensions.

## A.2 Astronomy

We utilize the AutoSciDACT pipeline to identify anomalous gravitational-wave sources. Our dataset comprises time-series recorded by the two advanced LIGO detectors [54] - Hanford, Washington, and Livingston, Louisiana - spanning the third observing run (O3) from April 2019 to March 2020; this data is publicly available [56, 57]. Our data preparation and labeling process closely follows the setup described in [46, 70]. Class-balanced reference sets are constructed by injecting simulated signals into real background; the background class is simply segments with no injections. We inject compact-binary coalescences including binary black hole mergers from phenomenological model IMRPhenomPv2 [79, 80] , (sine-) Gaussian signals [81], signals from cusps [82], kinks [83], double kink events [84], and signals from band-limited white noise bursts (WNBs) [85]. Moreover, a dedicated "glitch" class is obtained using Omicron's veto criteria [86].

Each input consists of two sequences, one for each detector, with each sequence containing 200 points, corresponding to a 50 ms long time series sampled at 4096 Hz. Exemplary time-series can be

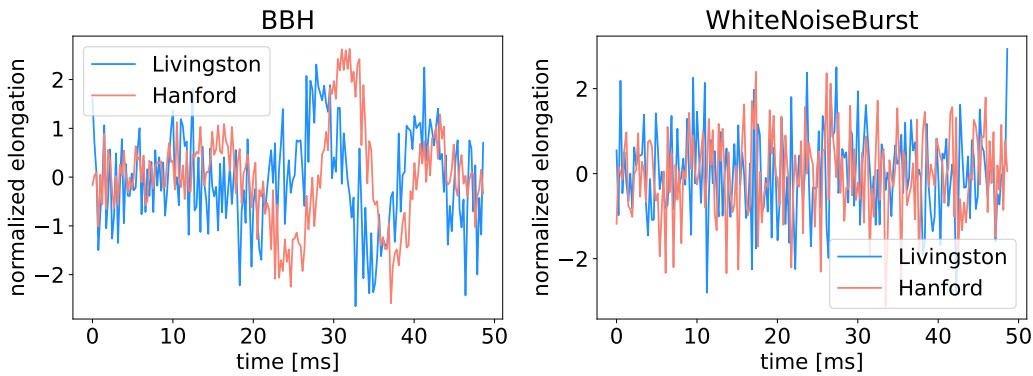

Figure 5: Example LIGO signal waveforms: Signals from binary black holes (left) and white noise burst (right).

found in Fig. 5. The dataset comprises a total of about 530,000 samples, with an near-uniform class balance in the pre-training. To enhance data processing and support network training, the data are normalized to have a unit standard deviation on a per-sample basis. Of the nine total classes, the WNB class is withheld from pre-training and treated as the held-out anomaly for discovery. White-noise bursts are deliberately model-agnostic: they represent correlated, band-limited stochastic fluctuations whose spectra are flat over the analysis band. As such they emulate the "worst-case" burst—lacking distinctive phase evolution or chirp structure, providing a stringent test of the pipeline's capacity to disentangle subtle, structure-poor signals from detector noise.

The available data is split into a training, validation and test dataset with the test dataset not only utilized for testing the pre-training performance, but also for constructing the reference $\mathcal{R}$ and data distribution $\mathcal{D}$ for the hypothesis test.

The optimization of the backbone encoder $f_\theta$ - a one-dimensional ResNet with about 7.2M trainable weights - uses the combined loss objective (SimCLR temperature $\tau = 0.5$, $\lambda_{CE} = 0.5$) and the AdamW optimizer [87] with an initial learning rate of 0.001 and 350 batch size. To facilitate improved convergence and generalization, a cosine annealing learning rate schedule is employed. The training is set up for a maximum of 25 epochs, with early stopping in case the validation loss does not decrease for more than five epochs.

### A.3 Particle Physics

JetClass[5] is an open-source particle physics dataset introduced in [60]. The training set consists of 100M jets from 10 classes (10M per class), of which we use five for a total of 50M training samples: QCD (quark/gluon), top quark ($t \rightarrow bqq'$), $W$ boson ($W \rightarrow qq'$), $Z$ boson ($Z \rightarrow q\bar{q}$) and Higgs ($H \rightarrow b\bar{b}$). The validation set consists of 500,000 jets per class and is used during training to monitor performance. The test set consists of 2M jets per class and is used to construct reference and data samples for all NPLM hypothesis tests presented in the main text.

For the encoder, we use the particle transformer architecture nearly exactly as described in [60], using a particle embedding of dimension 128, eight self-attention layers with eight heads, and two class-attention layers with the final 4-dimensional embedding derived from the final class token plus a fully connected layer. We use the same 17 per-particle input features described in [60], including information on the particle's energy/momentum, trajectory, and particle type. We cap the input size at 64 particles per jet. We train the encoder with a SimCLR temperature $\tau = 0.1$ and a classifier strength of $\lambda_{CE} = 0.1$, running for 100 epochs with an initial learning rate of $5 \times 10^{-4}$ annealed to $10^{-5}$ on a cosine schedule and using the AdamW optimizer [88]. We use a batch size of 512.

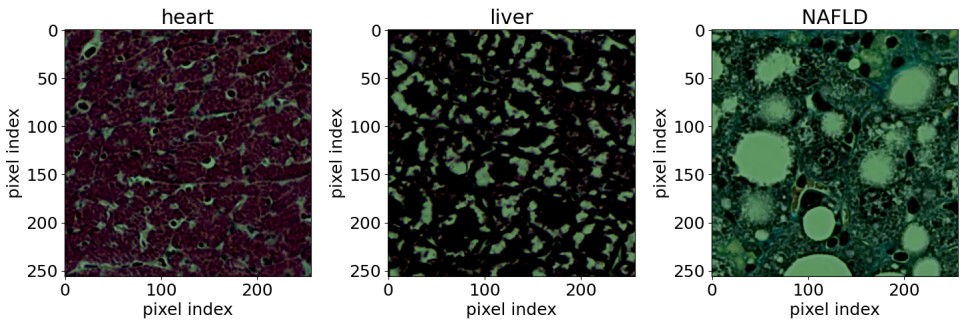

Figure 6: From left to right: exemplary image patches of heart tissue, liver tissue and liver tissue with a non-alcoholic fat Non-Alcoholic Fatty Liver Disease (NAFLD) [63].

## A.4 Histology

Exemplary image patches from the publicly available histological dataset [63] are shown in Fig. 6. The training dataset is balanced and includes samples from mouse tissues of the brain, heart, kidney, liver, lung, pancreas, and spleen and a separate class containing normal liver tissue samples from rats. Each class comprises approximately 6,300 samples. The dataset is split into training, validation, and testing sets with a ratio of 70%/20%/10%. In addition, an independent test set consists of approximately 2,300 samples of normal mouse liver tissue and an equal number of samples with non-alcoholic fatty liver disease (NAFLD).

Due to statistical constraints, the reference distribution $\mathcal{R}$ is constructed from the training data, with prior validation to ensure the absence of overfitting. The data distribution $\mathcal{D}$ for normal tissue is derived from both the training set and the independent dataset containing only healthy mouse liver samples.

The backbone encoder $f_\theta$, EfficientNet-B0 [64] with approximately 4.8M trainable parameters, is optimized using a combined loss objective, incorporating SimCLR contrastive loss (temperature $\tau = 0.5$) and cross-entropy loss with weighting $\lambda_{CE} = 0.5$. Optimization is performed with the AdamW optimizer, using an initial learning rate of 0.001 and a batch size of 32. To promote stable convergence and improved generalization, a cosine annealing learning rate schedule is employed. Training is conducted for a maximum of 25 epochs, with early stopping triggered if the validation loss fails to improve for more than five consecutive epochs. Each training run takes between one and two hours on a single NVIDIA A100 GPU.

## A.5 Images

We use the CIFAR-10 [65] dataset, applying the standard resize to $232 \times 232$ with interpolation, crop to $224 \times 224$, and standardizing using the ImageNet [89] mean and standard deviation. For the encoder, we use the Pytorch-provided pre-trained ResNet-50 [67] weights and replace the final fully connected layer with an MLP of hidden dimensions $[512, 256, 128]$ and an output dimension of 4. Only these final MLP weights are floated during training.

We use the 50,000 CIFAR-10 training images to pre-train the encoder, using only 45,000 in practice because class 1 is held out as the anomaly. When evaluating with NPLM we introduce 100,000 images from CIFAR-5m [66] in order to boost the number of points available for demonstrating our method. The approximately 5 million images in CIFAR-5m were generated by an unconditional denoising diffusion probabilistic model (DDPM) [90] trained on CIFAR-10, then labeled by the 98.5% accurate Big-Transfer model [91]. Trainings took about 1 hour on a single A100 GPU, and ran for 50 epochs with a learning rate of $10^{-3}$ annealed to $10^{-5}$ on a cosine schedule and with a batch size of 512. The SimCLR temperature is set to $\tau = 0.1$ and the cross-entropy strength to $\lambda_{CE} = 0.5$.

---

[5] https://zenodo.org/records/6619768

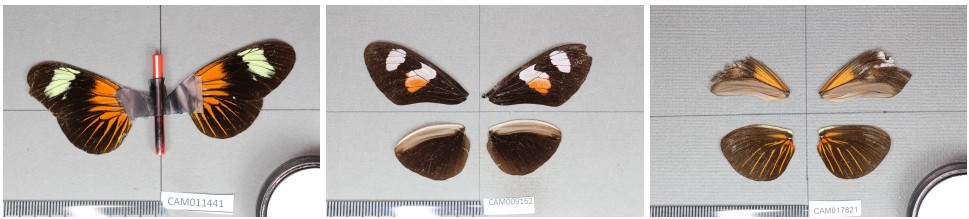

Figure 7: From left to right: Exemplary images of two different butterfly subspecies (left, center) and a hybrid offspring of those species (right).

## A.6 Genomics

As an example from genomics, we evaluate our approach on the dataset provided in the "Butterfly Hybrid Detection" challenge[6]. Butterflies come in different subspecies characterized by visual differences in color and pattern on the wings and can sometimes mix and produce offspring. We aim to detect these anomalies, so called hybrids, with AutoSciDACT.

The dataset consists of 1991 colour images of size $224 \times 224$ pixels, annotated to belong to one of 14 classes, with a highly imbalanced class distribution (some classes contain almost 100 times more instances than others) [92–115]. Example images of two different species and a hyrbid are shown in Fig.7. We adopt a fixed split of $80\%$ training, $10\%$ validation and $10\%$ test sets. Our model uses a backbone encoder based on the BioClip architecture [116], with approximately 430k trainable weights. Training employs the AutoSciDACT combined loss: a contrastive self-supervised term following the SimCLR formulation with temperature parameter $\tau = 0.5$, and a supervised cross-entropy term, weighted with $\lambda_{CE} = 0.5$. Optimization is performed using the AdamW optimizer at a learning rate of 0.001 and batch size of 32. We apply cosine annealing learning-rate scheduling over 100 epochs and minimal value 0.0001. Training is terminated by early stopping when no improvement is observed for five consecutive epochs; in our experiments this is the case after 43 epochs.

The contrastive embedding resulting from the projection of image data into a four-dimensional latent space is shown in Fig. 8 (left). Among all experiments, the anomaly in the genomics case shows the clearest separation from the background classes. The performance of AutoSciDACT in this setting is presented in Fig. 8 (right), which displays the statistical significance (Z-scores) obtained by NPLM and various baseline methods for detecting varying proportions of hybrid butterfly species images injected in background-dominated samples. We omit asymptotic results, since the underlying assumption that the test statistic is $\chi^2$-distributed, is invalid in light of the limited statistics of the dataset. The limited statistics of this dataset is also the reason why it is not included in the main results. Notably, the empirical Mahalanobis distance baseline achieves the strongest performance due to the excellent separation of the anomalous class in the contrastive embedding space. Conversely, the Maximum Mean Discrepancy (MMD) test yields the weakest performance, potentially due to suboptimal kernel width choice (see Appendix B.2 for further discussion).

## B  Additional Experiments

### B.1  Impact of NPLM kernel width

The choice of NPLM kernel width has a significant impact on the sensitivity of the test, and there is no *a priori* choice that can optimize sensitivity to potentially anomalous features. The results in Fig. 3 included Z-scores from the best-performing kernel and the average over all kernel widths considered. For completeness, we plot Z-scores from all kernel width settings in Figures 9 (asymptotic) and 10 (empirical). These figures exactly mirror Fig. 3, displaying results for each of the five datasets in the five panels (a)-(e). In Fig. 10 we also show the kernel-averaged $Z$-score in black. In general, intermediate to larger kernels do the best, with only fairly small variations among the top performers. The empirical Z-scores cannot exceed roughly $2.88\sigma$ due to the number of pseudo-experiments (500).

---

[6]https://www.codabench.org/competitions/3764/

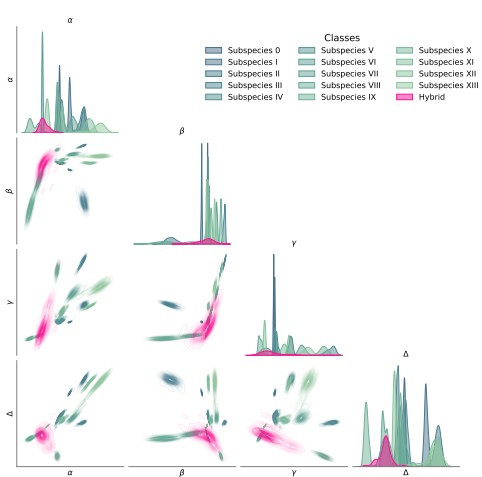
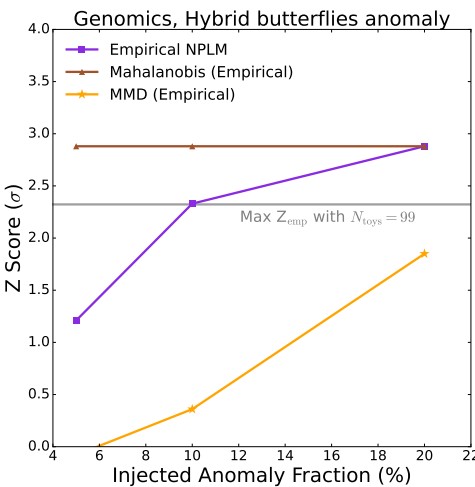

Figure 8: Results for the genomics experiment: Contrastive embeddings in four dimensions (left). Background classes are shown in hues of blue and green, while the anomaly is overlaid in hot pink. Results of the statistical tests (right).

## B.2 Comparisons with additional baselines

We further contextualize AutoSciDACT's performance by comparing with two additional anomaly detection baselines: Maximum Mean Discrepancy (MMD) [29] and Fréchet Inception Distance (FID) [117, 118].

**Nyström approximated Maximum Mean Discrepancy (N-MMD).** The Maximum Mean Discrepancy (MMD) is a kernel-based statistical test used to determine whether two datasets come from the same distribution [29]. It works by mapping the data into a high-dimensional reproducing kernel Hilbert space (RKHS) via a chosen kernel (e.g. Gaussian) and computing the distance between the mean embeddings of the two distributions in that space. MMD has been shown to be sensitive to subtle differences between distributions, making it particularly effective in high-dimensional settings. However, its computational cost, quadratic in the number of examples, limits its applicability to small sample sizes or low-dimensional problems. To address this, [119] introduced a scalable variant of the MMD test based on a Nyström approximation of the kernel matrix, enabling its use on larger datasets while maintaining statistical power. Since the NPLM test employed in this work is also built upon the Nyström approximation, we perform a direct comparison between Nyström-MMD and NPLM under matched settings, using the same number of centroids and same kernel bandwidth.

**Fréchet distance (FD).** The Fréchet distance has emerged as a popular metric for comparing probability distributions, particularly in the context of generative modeling [117, 118]. Under the assumption that both distributions are Gaussian, it admits a closed-form expression involving only their means and covariances, making it computationally efficient and interpretable. However, this assumption can limit its effectiveness when the underlying distributions exhibit significant non-Gaussian behavior, such as heavy tails or multimodality. Despite this, the Fréchet distance remains widely used due to its robustness in high-dimensional settings and its ability to capture both mean and covariance differences.

In Fig 11 we reproduce Fig. 3 with empirical MMD and FID Z-scores included for the particle physics, astronomy, histology, and image datasets. NPLM outperforms FID/MMD for the particle physics and image datasets, approximately matches them for astronomy, and, surprisingly, underperforms in histology. This wide range of outcomes makes it difficult to draw unambiguous conclusions, but broadly suggests that the "best" anomaly detection method depends strongly on the structure of the embedding space and the size of the dataset at hand. NPLM's sensitivity scales with available sample size, and the histology dataset has by far the smallest available test datasets among all experiments in the main body of the paper (see Table 1). In such data-constrained cases, other methods may perform better as they do not require likelihood ratio estimation. In other cases, the structure of the data embeddings may also confer an advantage (e.g. if clusters are approximately Gaussian). However,

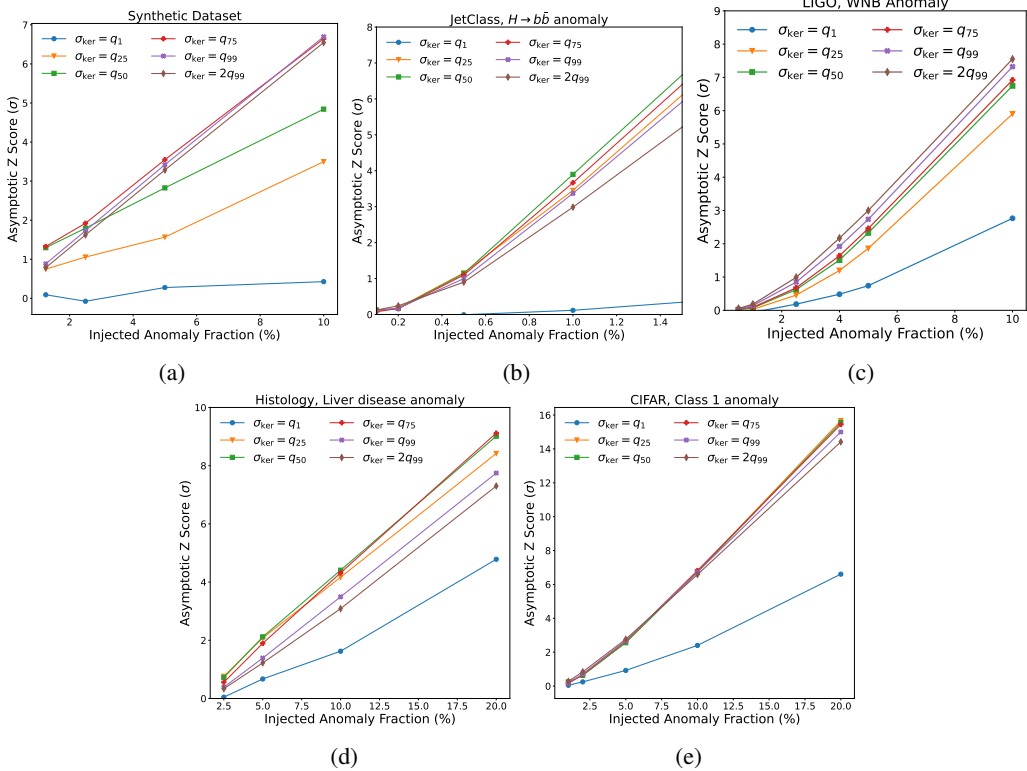

Figure 9: Asymptotic NPLM $Z$ scores as a function of injected signal yield for all six kernel choices $\sigma = [0.1, 1.5, 2.6, 3.6, 4.9, 9.8]$ used in pseudo experiments. We show results for our five benchmark datasets: Synthetic (a), JetClass (b), LIGO (c), Histology (d), and CIFAR-10 (e).

NPLM's strong performance in all but the most data-constrained cases positions it as a reliable and competitive choice.

### B.3 Varying embedding dimension

We use a very low contrastive embedding dimension ($d = 4$) for the main results of this paper. This choice was made to ensure the tractability of our statistical anomaly detection technique (NPLM), which relies on statistical hypothesis testing that can quickly lose sensitivity in high dimensions. In this section we explore the impact of increasing the embedding dimension, extending our experiments up to $d = 32$[7]. Figure 12 shows the Z-score as a function of $d$ for the CIFAR, JetClass, and LIGO datasets, where the best-performing kernel widths are used and the signal injection fractions are set such that NPLM has good but not fully-saturated sensitivity at the default setting $d = 4$.

There are no unambiguous trends for any of the anomaly detection methods, with all methods performing relatively stably up to $d = 32$. NPLM's sensitivity declines very modestly in the CIFAR and JetClass examples, but slightly *improves* in the LIGO example. The MMD and Fréchet metrics are similarly stable. Interestingly, the Mahalanobis distance appears to almost always benefit from a larger dimensionality. This could be explained by class-specific clusters having more "room" to spread out and condense under the contrastive objective in a higher-dimensional space. As Mahalanobis distance is sensitive to the distribution of these clusters, this would likely improve performance (assuming that the anomalous cluster is separated from the rest, i.e. not an overdensity in an existing cluster). The absence of clear trends for NPLM is encouraging, suggesting AutoSciDACT could be applied to problems requiring higher-dimensional latent spaces. Where possible, however, we vouch for keeping the dimensionality low as this is beneficial for classical statistical analysis and uncertainty quantification.

---

[7]This is still modest relative to standard embedding sizes in e.g. computer vision or natural language, but a reasonable choice for many scientific applications.

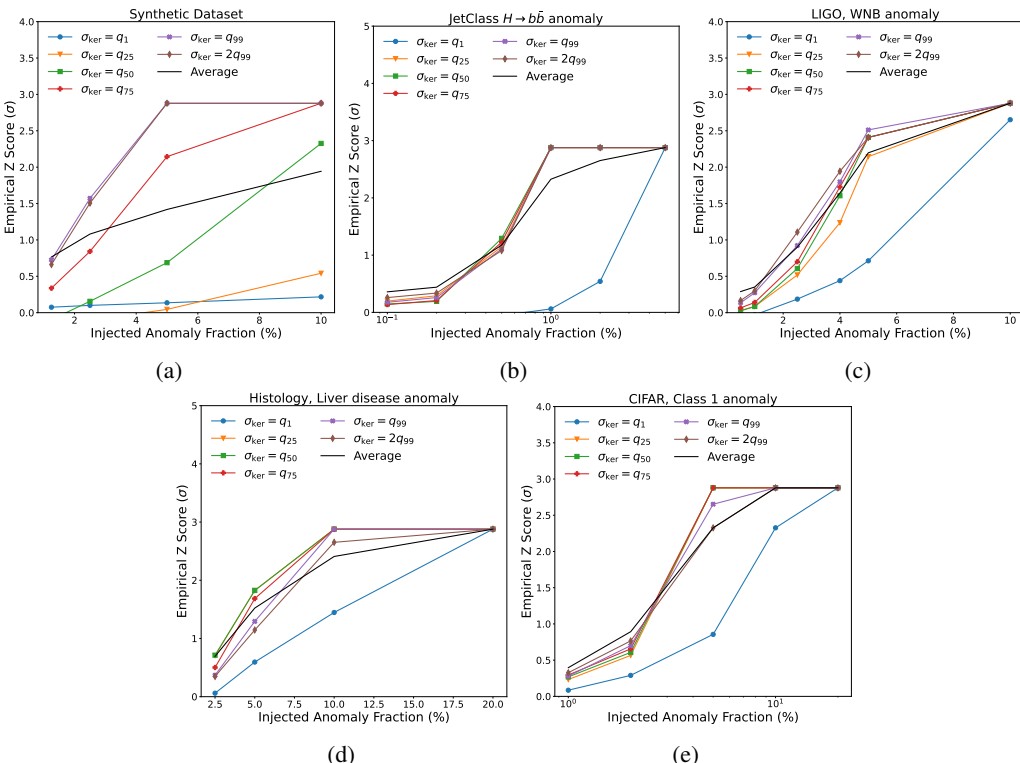

Figure 10: Empirical NPLM $Z$ scores as a function of injected signal yield for all six kernel choices $\sigma = [0.1, 1.5, 2.6, 3.6, 4.9, 9.8]$ used in pseudo experiments. We show results for our five benchmark datasets: Synthetic (a), JetClass (b), LIGO (c), Histology (d), and CIFAR-10 (e). We also show the average empirical $Z$ score from all six kernels in black.

## B.4 Noisy labels

Many real-world scientific datasets are labeled by hand or by algorithms with known error rates, both of which result in some fraction of mislabeled training data. To assess our sensitivity to such cases, we train variants of our JetClass contrastive embedding model with 1, 2, 5, and 10% label noise, meaning $x\%$ of samples the training datasets are randomly mislabeled. The results are shown in Fig. 13, where again the best-performing kernel width is chosen and the injected signal fraction is set such that NPLM has good performance at zero noise. As expected, sensitivity drops as label noise is increased, falling from $4\sigma$ to about $2\sigma$ at 10% noise.

## C Comparison to previous results from histology

Our setup for the histology study makes use of the same dataset as in [25]. In [25], the authors proposed a contrastive learning method for anomaly detection. The learned embedding returns a 320-dimensional representation, and a standard one-class SVM with the Radial Basis Function (RBF) kernel and margin error $\nu = 0.1$ is then trained on anomaly-free data to construct an anomaly score. Applying a threshold that ensures zero false positives on the training set, the tiles constituting a data sample are tagged as either standard or anomalous. The final anomaly metric used for comparisons is an average across the tags of all tiles in the sample. This statistical test assumes the anomaly is out-of-distribution and the distribution of the background class in the chosen representation is well clustered. While these assumptions are met in [25], out-of-distribution is not ensured for an arbitrary representation space agnostic to the anomaly source, which motivates our choice for using NPLM as a universal statistical test.

Since [25] does not present results in terms of $p$-value, we implement the one-class SVM trained on our four-dimensional embedding space for comparison purposes. We find that the performance

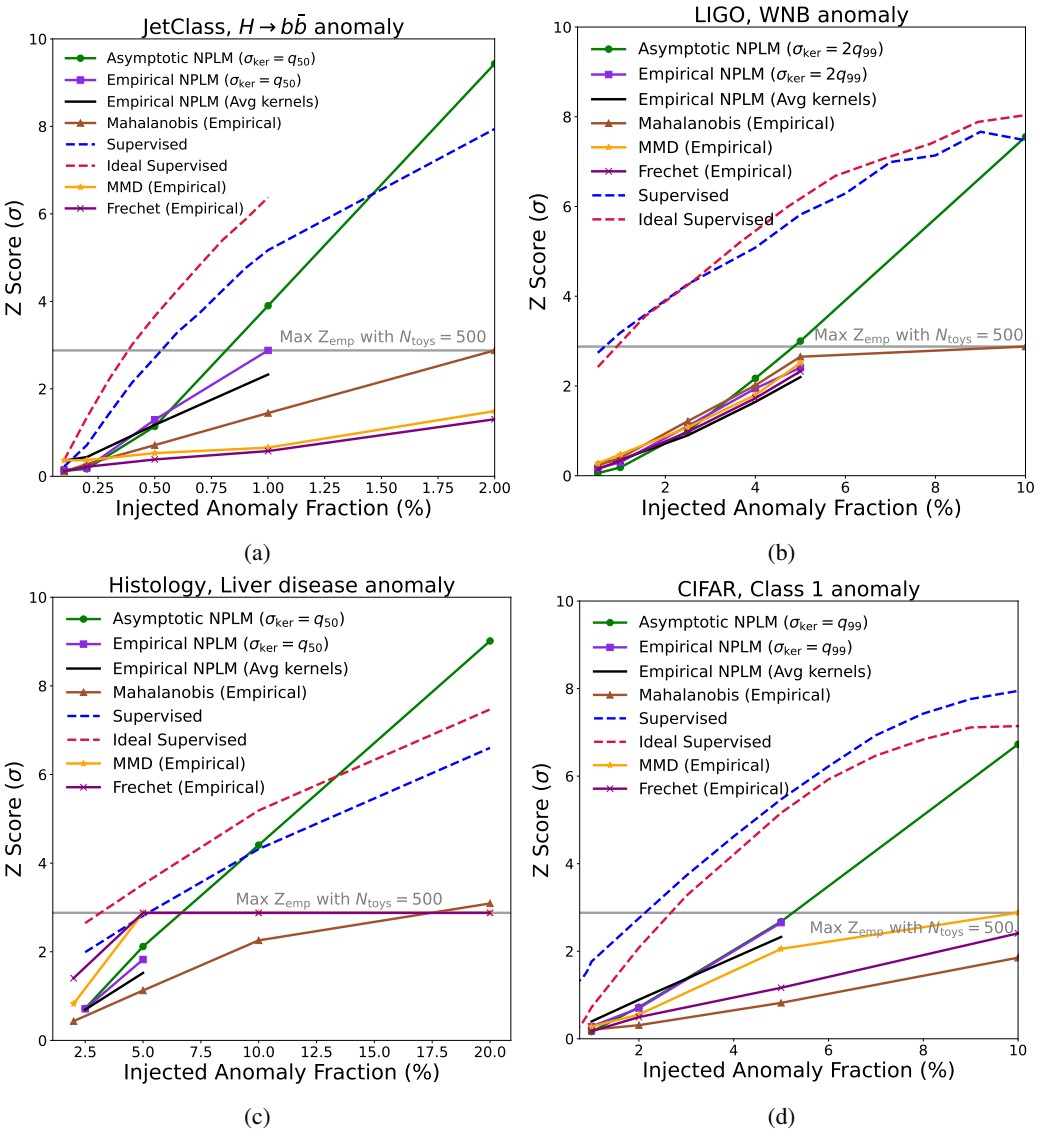

Figure 11: A reproduction of Fig. 3 including the Maximum Mean Discrepancy (MMD) and Fréchet Inception Distance (FID) baselines for the JetClass (a), LIGO (b), Histology (c), and CIFAR-10 (d) datasets.

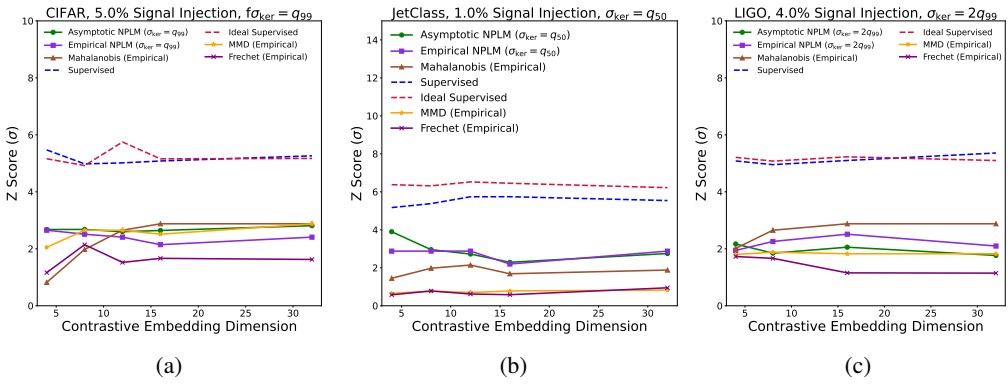

(a)                 (b)                 (c)

Figure 12: Z score as a function of embedding dimension for NPLM and baseline anomaly detection methods in the CIFAR-10 (a), JetClass (b), and LIGO (c) datasets. The best-performing kernel widths are chosen for presentation, and signal injection fractions are set such that NPLM has good but not fully-saturated sensitivity at the $d = 4$ baseline.

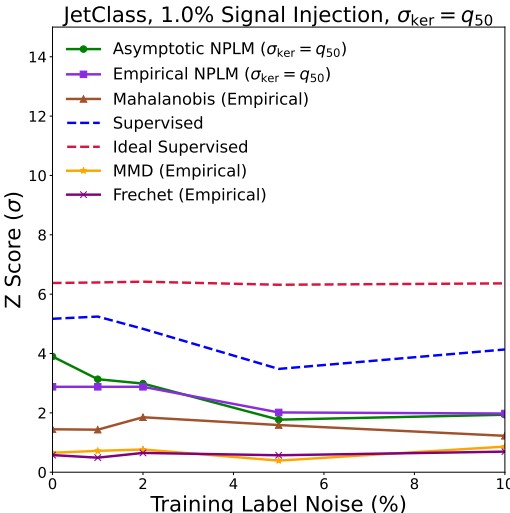

Figure 13: Z score as a function of label noise (%) dimension for NPLM and baseline anomaly detection methods for the JetClass dataset. The best-performing kernel widths are chosen for presentation, and signal injection fractions are set such that NPLM has good but not fully-saturated sensitivity at the zero-noise baseline.

of the test highly depends on the threshold set on the one-class output score. With a threshold that allows for 10% false positive rate, the one-class SVM performs comparably to the NPLM version. The one-class SVM has no discriminative power when the threshold is set to 1% false positive rate, corresponding to a more subtle anomalous contribution. The reason for such variance is the fact that in the latent representation, the anomaly does not necessarily lie outside the distribution. Additional tests run on the astronomy data (where the signal highly overlaps with one of the background classes) show an even more striking example of detection failure.

In conclusion, these studies reassure our strategy of using the NPLM to compute a two-sample test rather than focusing on out-of-distribution detection because two-sample tests are sensitive to a wider range of anomalous behavior. This is particularly relevant for the histology data since more than one anomalous tile is expected per sample, and capturing collective behaviors enables the detection of more subtle anomalies, e.g., to detect diseases manifesting in tissue earlier.

# D   Discriminating CIFAR-10.1 and CIFAR-5m

To further demonstrate the capabilities of AutoSciDACT for detecting distributional shifts, we apply it to the problem of quantifying the difference between CIFAR-10 and the synthetic CIFAR-5m dataset [66], as well as the independently curated CIFAR-10.1 dataset [120]. CIFAR-5m was discussed in the main text and App. A, and consists of approximately 5 million images generated by a diffusion model trained on CIFAR-10. CIFAR-10.1 was introduced in [120], where it was specifically collected and curated to be as close to CIFAR-10 as possible (e.g. drawing from similar sources of images). The idea was to test whether classifiers trained on CIFAR-10 readily generalized to data outside of CIFAR-10 but in principle distributionally identical. It was found that performance dropped substantially when evaluating on CIFAR-10, indicating some non-trivial shift in the dataeset. Recent work in Liu et. al. [121] and Guille-Escuret et. al. [13] has applied some versions of two-sample tests to distinguish CIFAR-10 and CIFAR-10.1, both showing strong evidence of a distributional shift between the two.

In Fig. 14 we tackle this question with AutoSciDACT. We train a four-dimensional contrastive embedding on the full CIFAR-10 training dataset without holding out any classes, as individual classes are not considered anomalous in this context. We use the same architecture and training procedure as for the CIFAR-10 results in the main text. We use this encoder to embed the CIFAR-10m test set, the CIFAR-10.1 set, and 100,000 randomly selected images from the CIFAR-5m set with the same class proportions as CIFAR-10 test and CIFAR-10.1. We run 500 NPLM pseudo experiments for each of the six kernel widths $\sigma = [0.1, 1.5, 2.6, 3.6, 4.9, 9.8]$ to produce the following distributions of test statistics:

1. **Null hypothesis:** in each experiment $\mathcal{R}$ is composed of 8500 randomly sampled CIFAR-10 test set images, and $\mathcal{D}$ from the remaining 1500.

2. **CIFAR-10.1:** in each experiment $\mathcal{R}$ is composed of 8500 randomly sampled CIFAR-10 test set images, and $\mathcal{D}$ from 1500 randomly sampled CIFAR-10.1 images.

3. **CIFAR-5m:** in each experiment $\mathcal{R}$ is composed of 8500 randomly sampled CIFAR-10 test set images, and $\mathcal{D}$ from 1500 randomly sampled CIFAR-5m images.

Distributions of the NPLM test statistic for each scenario and each kernel width are plotted in Fig. 14, with the corresponding asymptotic and empirical $Z$ scores for CIFAR-10.1 and CIFAR-5m relative to the CIFAR-10-only null hypothesis indicated in the legends. Even the smallest and worst-performing kernel $\sigma = 0.1$ distinguishes CIFAR-10.1 from CIFAR-10 at the $2.2\sigma$ level, while the remaining larger kernels distinguish it extremely easy beyond even the $10\sigma$ level. This indicates a clear distributional shift between CIFAR-10 and CIFAR-10.1, and presents one of the first (to our knowledge) statistically rigorous quantifications of this discrepancy.

The discrepancy between CIFAR-10 and CIFAR-5m is notably *much* smaller, saturating near $2.3\sigma$ for the best performing kernels. This is exactly in line with what one would expect, given that CIFAR-5m is generated from a diffusion model trained on CIFAR-10. A well-trained diffusion model is able to model its training distribution exceedingly accurately, and the relatively small deviation we observe here underscores this fact. More interestingly, this hints at an unexpected but fascinating potential use case for AutoSciDACT as a method for evaluating the quality of generative models.

# E   Searching for the Higgs boson in LHC Data

To demonstrate how AutoSciDACT might be used in a more realistic setting, we use it to search for evidence of the Higgs boson ($H$) in a dataset of real proton-proton collision data collected by the Compact Muon Solenoid (CMS) experiment at the Large Hadron Collider (LHC). We specifically target the four-lepton final-state, where the Higgs boson decays to four electrons, four muons, or two electrons and two muons ($pp \rightarrow H \rightarrow e^+e^-e^+e^-$, $\mu^+\mu^-\mu^+\mu^-$, or $\mu^+\mu^-e^+e^-$). The Higgs boson was discovered through the observation of an excess in predominantly these final states and the di-photon final state [122, 123], with Higgs-like events isolated using hand-tuned selections on physics-motivated variables that were reconstructed from observed data. Here, we replace the majority of this selection with the AutoSciDACT pipeline, keeping only a loose "pre-selection" of events designed to suppress large, well-known background processes.

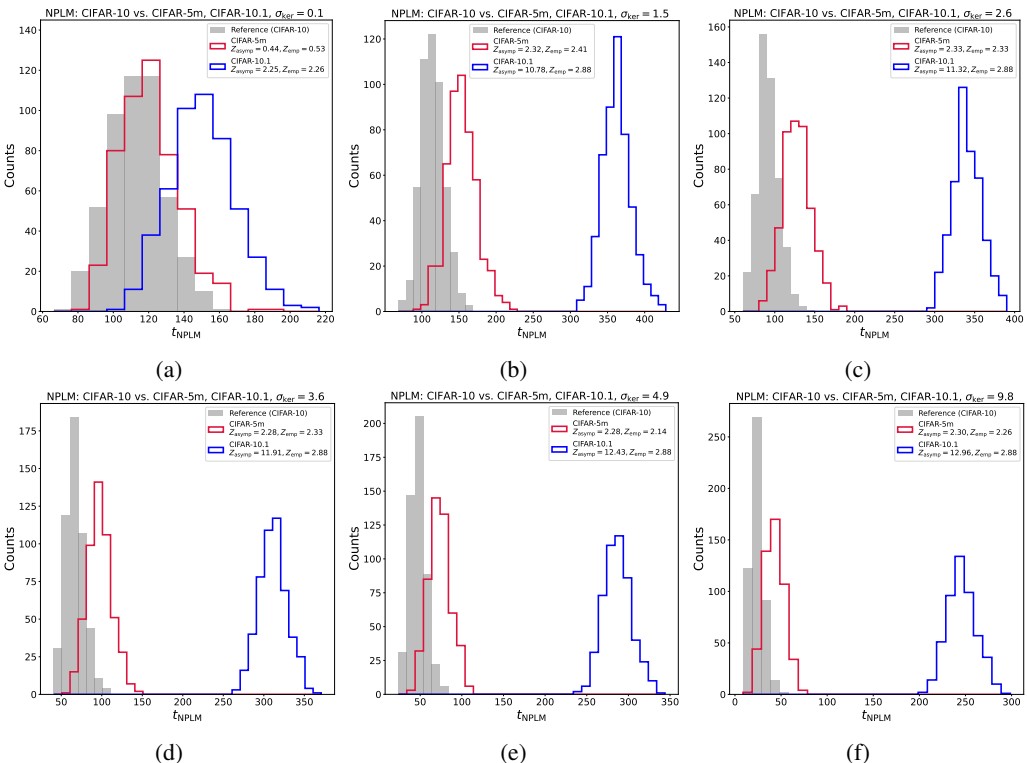

Figure 14: NPLM tests comparing CIFAR-10 to CIFAR-5m and CIFAR-10.1 using a four dimensional contrastive embedding. The reference is built from the CIFAR-10 test set, and test stastics for the null hypothesis are shown in gray. Test statistics comparing against CIFAR-5m are shown in red, while those comparing against CIFAR-10.1 are shown in blue. Panels (a)-(f) corresponding to the six different NPLM kernel widths $\sigma = [0.1, 1.5, 2.6, 3.6, 4.9, 9.8]$.

We run AutoSciDACT on both real and simulated events from the CMS Open Data [124] for 2011 and 2012. We impose a similar data pre-selection to the standard CMS search, requiring at least 4 well-identified and loosely isolated electrons or muons with transverse momenta ($p_T$) greater than 5 GeV, the most energetic of which ("leading lepton") having $p_T > 20$ GeV. Additionally, we require one opposite-sign pair of electrons or muons with invariant mass ($m_{\ell^+ \ell^-}$) greater than 12 GeV. In the first stage of AutoSciDACT, we pre-train the contrastive encoder using simulated events from the dominant background processes ($Z$ boson pair production with decays to the $4e$, $4\mu$, and $2e2\mu$ final states). The encoder is a small MLP that takes as input the full kinematic information ($p_x, p_y, p_z, E$) and particle ID (electron or muon) of the four leptons in the event, for a total of 20 inputs. As in the main paper, we train a four-dimensional embedding space.

In the search phase, we follow the standard procedure for CMS data analysis and compute expected $p$-values based on simulated data, then measure the observed $p$-value in real data. Expected $p$-values are computed empirically by running many toys for the background-only and background + Higgs hypotheses to obtain distributions of a test statistic (either the NPLM test statistic or one obtained from a direct fit to the four-lepton invariant mass, see below). For each NPLM toy, we sample the reference ($\mathcal{R}$) and observed ($\mathcal{D}$) datasets from simulated background events, adding additional simulated Higgs events to $\mathcal{D}$ for toys under the background + signal hypothesis. Finally, a single test statistic is computed using the *true* data (i.e. observed in CMS), and the observed $p$-value is computed relative to the test statistic for background-only toys.

We perform three different tests using the statistical procedure described above:

- **Baseline Discovery** We assume full knowledge of the Higgs boson, and perform a one-dimensional hypothesis test using the single most discriminating variable: the four-lepton invariant mass $m_{4\ell}$. The Higgs boson signal should manifest itself as a localized "bump" or peak in a histogram of observed $m_{4\ell}$ at $m_{4\ell} = m_H$ (approximately 125 GeV). We construct

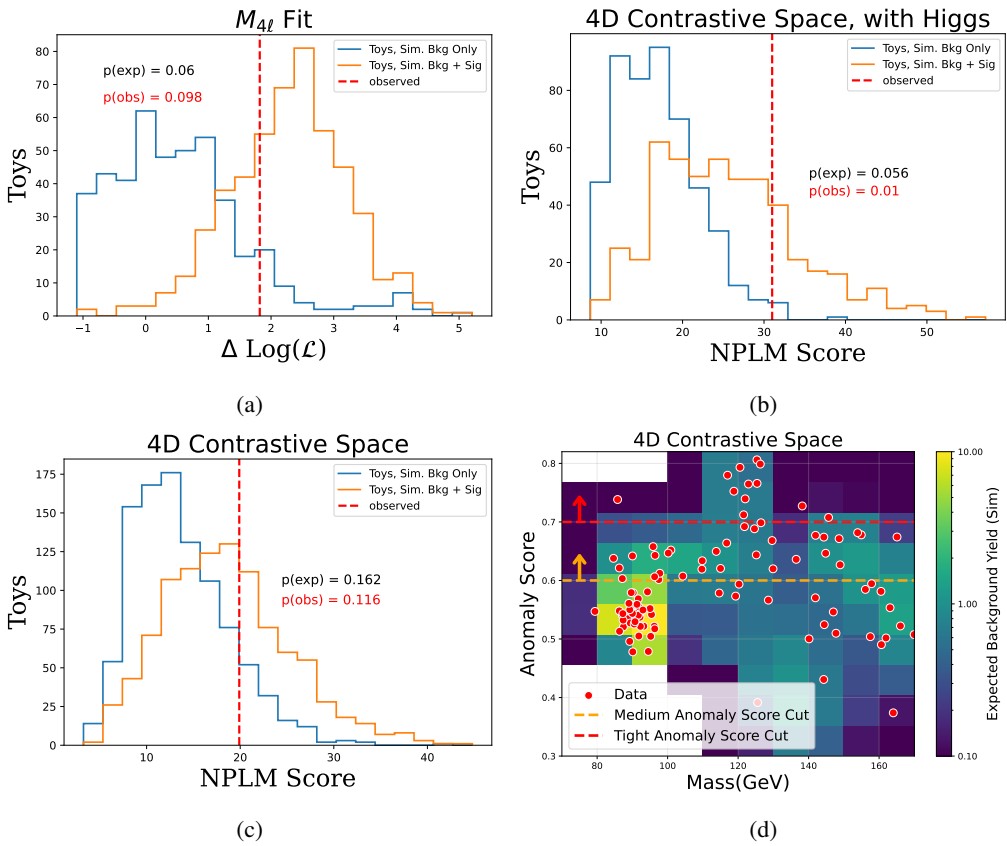

Figure 15: Top row: test-statistic distributions for simulated background-only (blue) (orange) simulated background + Higgs signal (orange), along with observed test statistic (red), for a direct fit to $m_{4\ell}$ (left) and AutoSciDACT with Higgs included in training (right). Bottom left: Expected and observed test statistics/$p$-values for vanilla AutoSciDACT trained without knowledge of the Higgs. Bottom right: Distributions of $m_{4\ell}$ versus AutoSciDACT NPLM anomaly score in simulated backgrounds (filled) and observed data (points), where the most anomalous observed data lies near the true Higgs mass (125 GeV).

background-only ($B$) and signal ($S$) $m_{4\ell}$ histogram shape templates using simulated events, then fit the observed $m_{4\ell}$ histogram to the background-only ($B$) or signal + background ($S + B$) hypotheses. The statistical significance of an observation is then computed from the delta log-likelihood test statistic obtained from the two fits: $\Delta \log \mathcal{L} = \log \mathcal{L}_{S+B} - \log \mathcal{L}_B$. We view this as the baseline sensitivity for discovery.

- **AutoSciDACT NPLM** We run NPLM on the four-dimensional embedding space obtained from the contrastive encoder. We expect the presence of domain shift between the simulated and observed data (i.e. mis-modeling) to somewhat weaken the sensitivity of discovery.

- **Supervised AutoSciDACT NPLM** We re-train our contrastive encoder with simulated Higgs events included in the training set, so that it explicitly learns about the signal of interest. The Higgs signal should be well-separated from backgrounds in the learned embedding space, so we expect sensitivity in this case to be on par with the baseline obtained from the $m_{4\ell}$ fit.

We present results from our analysis in Fig. 15. The top row shows expected test statistic distributions and expected/observed $p$-values for the baseline (left) and *supervised* AutoSciDACT (right) methods. Both methods achieve similar expected performance, with the observed $p$-values differing somewhat in either direction due to known domain shifts between simulated and real LHC data. These shifts lead to a larger spread in observed $p$-values, and given that we are not properly accounting for systematic uncertainties in our analysis, they have a noticeable effect. The expected $p$-values are thus more

informative, as they provide a clear impression of how effectively each method finds a Higgs signal without the impact of domain shift.

Figure 15(c) shows results using *standard* AutoSciDACT (i.e. a contrastive space trained without knowledge of the Higgs). As expected, it is somewhat less sensitive than the baseline methods. However, even the baselines do not achieve highly significant $p$-values, largely due to the small dataset size and the fact that we only consider one Higgs decay mode[8] To better interpret these results, in Fig. 15(d) we plot two-dimensional distributions of $m_{4\ell}$ versus NPLM anomaly score for simulated backgrounds (filled) and observed data (points). We observe that the most anomalous points in *real data* are clustered near 120-130 GeV, near the known mass of the Higgs (125 GeV). We draw two horizontal lines depicting example medium/tight thresholds on the NPLM score, showing how these Higgs-like events could be isolated in a dataset by selecting on anomaly score. Note that a full analysis using this selection would likely increase sensitivity, since it would use both the 4-lepton mass and the anomaly score.

## F    Embedding Space Visualizations

For reference, we include visualizations of the four-dimensional contrastive embedding spaces for the CIFAR-10, JetClass, LIGO, and histology datasets in Figure 16.

---

[8]The expected significance in this channel was only $\sim 2\sigma$ in the original CMS paper [122], with discovery claimed only by combining results from several channels.

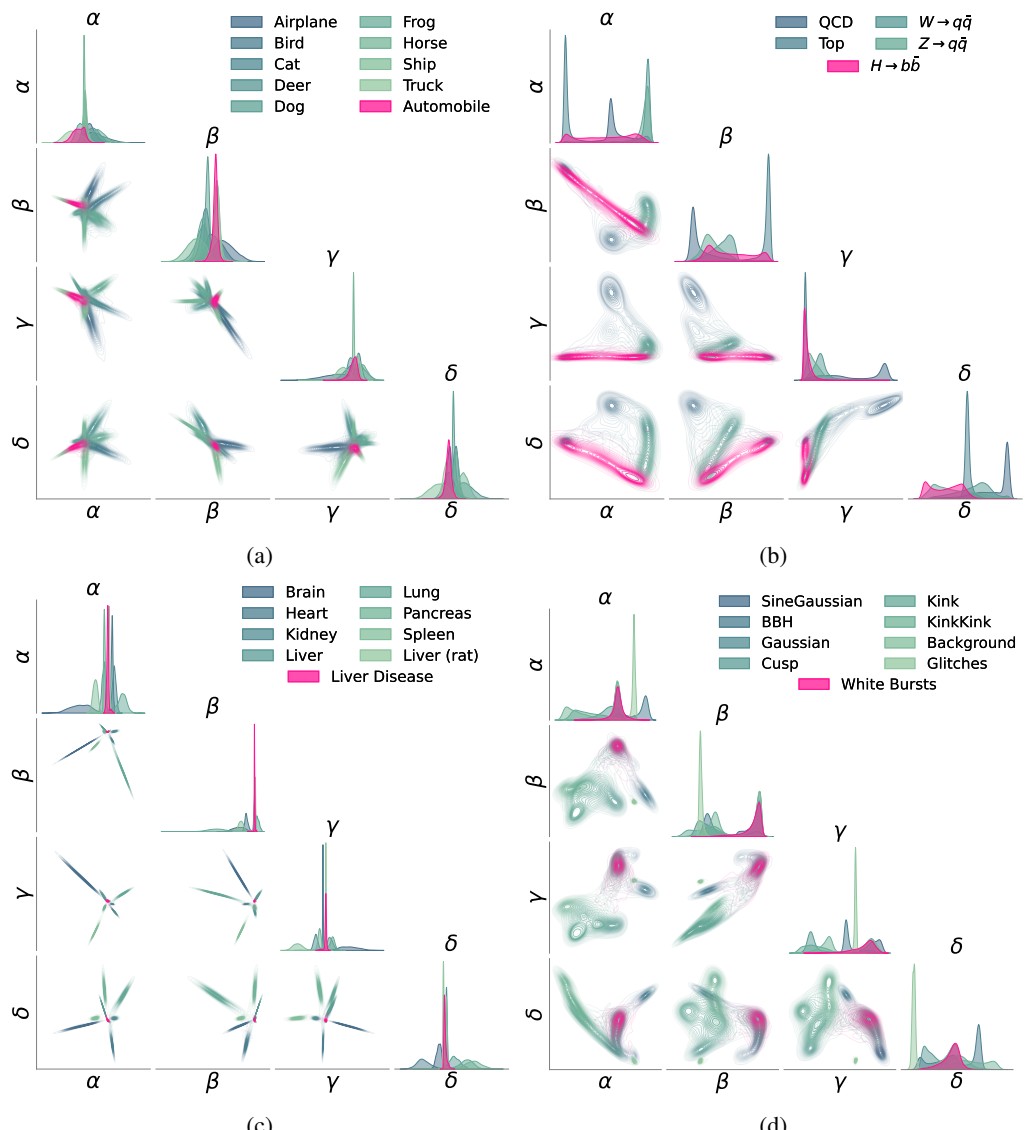

Figure 16: Corner plots showing the four-dimensional contrastive embedding spaces for CIFAR-10 (a), JetClass (b), histology (c), and LIGO (d). The turqoise clusters correspond to the classes used in training the encoder, and the pink cluster shows the distribution of "anomalous" signal in the learned space.

