# OpenReview forum: "AutoSciDACT: Automated Scientific Discovery through Contrastive Embedding and Hypothesis Testing"
_NeurIPS.cc/2025/Conference — NeurIPS 2025 poster_

### Official Review · Reviewer_qPQa · 2025-06-26

**Clarity:** 3
**Significance:** 3
**Originality:** 2
**Rating:** 5
**Confidence:** 3

**Summary:**

This paper introduces AutoSciDACT, a two-stage pipeline for scientific discovery. The two stages are: (1) Constructing a low-dimensional embedding using contrastive learning, and (2) employing the New Physics Learning Machine (NPLM) framework, performing a likelihood ratio test comparing a background distribution to another dataset with unknown composition. Extensive experiments on data sets from four diverse application domains (and one synthetic data set) demonstrate the effectiveness of the approach.

**Questions:**

- Why are the supervised baselines considered an upper limit (see line 285)?
- Is it possible to compare the method to baselines like maximum mean discrepancy [1] or the Fréchet distance [2, 3]?
- What is the justification for using a single, tied variance for all classes instead of a full covariance matrix for the Mahalanobis distance baseline (like, e.g., in [4])?
- Have augmentations been used in the pre-training stage in the individual domains? If so, which augmentations?
- How was the embedding dimension $d$ selected? $d = 4$ seems rather small compared to SOTA models.

**References**

[1] Gretton et al., A Kernel Two-Sample Test, JMLR 2012

[2] Dowson et al., The Fréchet Distance between Multivariate Normal Distributions, Jounral of Multivariate Analysis 1982

[3] Heusel et al., GANs Trained by a Two Time-Scale Update Rule Converge to a Local Nash Equilibrium, NeurIPS 2017

[4] Lee et al., A Simple Unified Framework for Detecting Out-of-Distribution Samples and Adversarial Attacks, NeurIPS 2018

**Ethical Concerns:**

["NO or VERY MINOR ethics concerns only"]

**Final Justification:**

The authors have added additional baselines and their responses were satisfactory.

**Limitations:**

Yes

**Quality:**

4

**Strengths And Weaknesses:**

**Strengths**

- The paper is well-written. The process of scientific discovery establishes a good motivation for the paper.
- AutoSciDACT employs a statistical test for discovery, which allows the user to assess significance levels.
- The work includes a very extensive set of experiments on four diverse domains (astronomy, particle physics, histology, and images), showcasing the wide applicability of the pipeline.

**Weaknesses**

- The suggested pipeline is merely a combination of two existing methods (Contrastive Learning + NPLM), which diminishes the methodological contributions of the paper.
- The paper is very dense, making certain sections rather hard to follow (e.g., the section introducing the supervised baselines - at least a more detailed discussion in the Appendix would have been beneficial, in my opinion)
- The selection of baselines could be extended (see Questions).

---

> ### Author Rebuttal · Authors · 2025-07-31
>
> We thank the reviewer for their detailed and constructive feedback. We are pleased that they found the framing around the process of scientific discovery provides a strong and well-motivated foundation for our approach. We appreciate the recognition of AutoSciDACT’s broad applicability and domain generalization. We address each of the weaknesses, suggestions, and questions that the reviewer raised and offer clarifications. We believe that the revisions have significantly strengthened the manuscript.
>
> ***
>
> > **The suggested pipeline is merely a combination of two existing methods (Contrastive Learning + NPLM), which diminishes the methodological contributions of the paper.**
>
> We thank the reviewer for raising this concern. While the individual components of our approach may not be novel in isolation, we believe their combination is both original and impactful. It enables the application of statistical anomaly detection techniques to complex, high-dimensional, and structured datasets commonly encountered in scientific domains.
>
> Recent work on representation learning tends to favor high-dimensional embeddings to preserve rich semantic information. However, these high-dimensional spaces are often difficult to interpret and analyze statistically, posing challenges for uncertainty quantification and hypothesis testing, which are central in the scientific domain. The NPLM method and other statistical methods degrade in their ability to find statistically meaningful overdensities when random, non-discriminating variables are added to the dataset (see figure 4 in Appendix A).  Our contribution shows that contrastive learning, when guided by domain knowledge (i.e. class labels), can extract low-dimensional representations that retain sufficient information to enable anomaly detection in a statistically robust manner, removing information that is redundant or non-critical. Beyond anomaly detection, this approach demonstrates how previously developed statistical methods, widely trusted and used in the scientific community, can be successfully integrated with state-of-the-art machine learning techniques for representation learning. This opens new avenues for scientifically grounded, data-driven discovery.
>
> ***
>
> > **The paper is very dense, making certain sections rather hard to follow (e.g., the section introducing the supervised baselines)**
>
> Thank you for this feedback! We acknowledge that the paper is dense and agree that the section introducing the supervised baselines could be more explicit. Our goal was to concisely present multiple baselines within limited space, but we understand that this may have made the section difficult to follow.
>
> The goal of the supervised baselines is to identify "best case" performance for anomaly detection given the contrastive embeddings, and to compare this to our anomaly detection algorithm (NPLM). The distinction between "supervised" and "ideal supervised" is whether or not the *true signal* was included as a class in the supervised contrastive training. Including it creates a space that is designed with the signal in mind, allowing "ideal" separation from background. We will add these clarifications to the revised version of the paper.
>
> ***
>
> > **The selection of baselines could be extended**
> > **Is it possible to compare the method to baselines like maximum mean discrepancy or the Fréchet distance?**
>
> Thank you for these helpful suggestions! Following the reviewer’s suggestion, we have performed studies with two additional baselines: Maximum Mean Discrepancy (MMD) and Frechet Distance (FD). We have evaluated these baselines for the gravitational wave, particle physics, and CIFAR datasets. We use the “empirical” method to compute p-values/z-scores with these metrics, as their asymptotic distributions are unknown. The results vary somewhat between datasets, but in all cases the NPLM (our method) performs equally well or outperforms MMD/FD with the default $d = 4$ contrastive embedding space. MMD/FID and NPLM perform more equally at lower anomaly injection fractions, while NPLM becomes significantly better for larger injections. Interestingly, we do not observe consistent trends in the relative performance among the three non-NPLM methods (Mahalanobis, MMD, FID), other than that they tend to lag behind NPLM. This highlights the complementarity of the various statistical anomaly detection methods we compare with NPLM. **This study will appear in the updated camera-ready version of our paper, though unfortunately its results (i.e. plots) cannot be shared here.**
>
> > **Why are the supervised baselines considered an upper limit (see line 285)?**
>
> Thank you for highlighting this point and allowing us to clarify! The idea of the supervised baselines is to probe best-case performance for identifying signals *given the contrastive embedding*. They differ in whether or not the true signal was included in training the space ("ideal"), with "ideal" being close to the best low-dimensional representation one could have with a particular signal in mind. In contrast, the “supervised” baseline represents a more realistic intermediate case, where some signal information is used only at the inference stage. Since data compression is a core part of our pipeline, we argue that it is natural to baseline our methods in this compressed space.
>
> Comparing these two baselines also offers insight into the learned representations: if the “supervised” and “ideal supervised" match, it indicates that the representation learned through signal-agnostic supervised contrastive learning is well-suited for downstream anomaly detection, even without prior knowledge of the signal.  Additionally, comparing our approach directly to the “supervised” baseline quantifies the impact of lacking prior signal information on test performance. Finally, the “ideal supervised” case provides a meaningful reference point for assessing the intrinsic difficulty of the anomaly detection task itself. Notably, the signal injection rate alone does not fully capture the problem's complexity, as the detectability of rare perturbations also depends on the separation between signal and background in the embedding space.
>
> > **What is the justification for using a single, tied variance for all classes instead of a full covariance matrix for the Mahalanobis distance baseline (like, e.g., in [4])?**
>
> We thank the reviewer for spotting the difference and have adopted the computation as suggested in [4]. The text in the manuscript is changed to reflect the update. Switching to this version of the Mahalanobis distance had negligible impact on the results we obtain in our studies, and the manuscript has been updated with the new numbers.
>
> > **Have augmentations been used in the pre-training stage in the individual domains? If so, which augmentations?**
>
> We do not make use of any augmentations. We elaborate on this in Sec. 3.1, where we justify our choice of using labels with the richer domain knowledge they encode compared to augmentations, tailored view of inputs. Choosing augmentations also typically requires specific domain knowledge, and it may be difficult to design augmentations that cover as rich a variety of positive pairs as labels encode.
>
> While we do not use augmentations, they can be easily added to the contrastive embedding if desired, for example, in cases where labels are unavailable or to impose certain invariances in the embedding. The downstream anomaly detection and hypothesis testing part of AutoSciDACT is unaffected.
>
> > **How was the embedding dimension d selected? d=4 seems rather small compared to SOTA models.**
>
> This is a very important aspect of our method and we thank the referee for calling attention to it! In principle, the optimal choice of $d$ for some signal cannot be known without *a priori* knowledge of the signal. The choice of $d$ is instead guided by another crucial constraint: *statistical tractability*. Performing statistically robust anomaly detection and producing calibrated test statistics is very difficult in high dimensions, quickly requiring a prohibitive number of samples as $d$ grows. We choose $d$ to allow for a powerful and tractable test while retaining enough flexibility to learn a good contrastive embedding.
>
> To further address this point, we have performed additional studies exploring our pipeline’s performance with larger embedding dimensions $d = 8, 16, 32$, with all other factors (training time, NPLM kernel size, etc) kept the same as in the main results (Fig 3). At modest signal injections – where NPLM and any of the other baselines are able to detect an anomaly with $\sim2-3\sigma$ significance or higher – we observe fairly mild variation in NPLM’s performance up to $d = 32$. The sensitivity degrades slightly as dimension is increased, but by a nearly negligible amount (0 to 0.5$\sigma$) and not in all cases. The MMD and FD metrics are similarly stable as a function of dimension, varying only slightly and sometimes non-monotonically up to $d=32$. The Mahalanobis distance, on the other hand, tends to perform better with increasing dimensionality. We hypothesize that this is due to more dimensions helping the background clusters be pushed further apart under the SimCLR objective. Since Mahalanobis is defined relative to these cluster centroids, a collection of tighter/better-separated clusters would be more sensitive to outliers. **This study will appear in the updated camera-ready version of our paper, though unfortunately its results (i.e. plots) cannot be shared here.**
>
> ***
>
> **Summary**
> We thank the reviewer again for their thoughtful and constructive feedback. We hope that our responses have clarified key details of our work and addressed the reviewer's concerns. Please let us know if we can further clarify, explain, or expand upon any specific points - we would be quite happy to continue a dialog during the discussion period!

---

> > ### Comment · Reviewer_qPQa · 2025-08-01
> >
> > I thank the authors for the comments and clarifications. My concerns have been addressed, and I have increased my score to 5.

---

### Official Review · Reviewer_jLRS · 2025-07-01

**Clarity:** 2
**Significance:** 3
**Originality:** 3
**Rating:** 4
**Confidence:** 3

**Summary:**

The authors observed that in scientific discovery, novel findings are tied to data points that are not in the background distribution. Based on this observation, they utilize anomaly detection with contrastive learning, using huge amounts of good quality simulation data to find anomalous data patterns. With the help of the New Physics Learning Machine (NPLM), they can then test the statistical significance of these findings to determine if the data is truly out-of-distribution. This significance score serves as an important tool in scientific research. All in all, the pipeline is a proper use case of anomaly detection and contrastive learning to help scientific discovery.

**Questions:**

1. In Lines 201-202, the description of the synthetic dataset mentions rotating 'The full M × N dimension space.' This phrasing was slightly confusing, as my understanding is that N represents the number of distinct data clusters, while the data itself exists in a (D+M)-dimensional space. Could you please clarify if the text was intended to state that the full (D+M)-dimensional space is rotated to obscure the signal?

2. In Figure 2 (right panel), in the 2D projection, the anomalous signal class (pink) appears to have a significant visual overlap with one of the background classes (blue). This raises a question about the pipeline's effectiveness in this specific case. Could you elaborate on how detection is achieved here? For example, is the separation clearer in other, un-plotted dimensions of the embedding, or is the NPLM test sensitive enough to detect a subtle overdensity even within this overlapping region?

3. In Figure 3 (d) and (f), in these two panels, the 'supervised' baseline (blue line) consistently outperforms the 'ideal-supervised' baseline (red line), which is the opposite of the trend seen in the other datasets. It's notable that both of these are image-based tasks. Could you offer some insight into this counter-intuitive result?

**Ethical Concerns:**

["NO or VERY MINOR ethics concerns only"]

**Final Justification:**

The author provided very detailed reply adressed my concerns. I decided to raise my mark to 4.

**Limitations:**

The authors disscussed the limtations.

**Quality:**

3

**Strengths And Weaknesses:**

Strengths:
1. The paper's primary strength is its statistical rigor. The entire pipeline is grounded in a formal hypothesis testing framework that quantifies the significance of any detected novelty through p-values and Z-scores, providing a robust foundation for scientific claims.
2. A key contribution is the paper's insightful framing of scientific discovery as a statistical anomaly detection problem. This provides a clear and actionable paradigm for automating the search for novel phenomena in large datasets.

Weaknesses:
1. The empirical validation of the method is limited. Despite showing success on 3 tasks (exclude toy data and CIFAR-10), the applications are drawn exclusively from the domains of physics and astrophysics, which is insufficient to substantiate the paper's broad claims of general scientific utility. Demonstrating efficacy in disparate fields, such as genomics, chemistry, or climate science, would be essential for this.
2. The paper's conceptualization of "scientific discovery" is limited, which narrows the framework's scope and makes its title potentially misleading. The term "discovery" often implies not just the detection of a new phenomenon, but also the generation of an explanatory hypothesis for it. The AutoSciDACT framework is tailored for the first step—identifying observational novelty—but its scope does not extend to the crucial subsequent step of building models or inferring the causal mechanisms that would explain the detected anomaly.

---

> ### Author Rebuttal · Authors · 2025-07-31
>
> We thank the reviewer for their detailed review and thoughtful feedback! We particularly appreciate their acknowledgment of our paper’s core contribution: a statistically rigorous treatment of anomaly detection, which makes it compatible with scientific hypothesis testing and robust claims of discovery. We are also pleased that our pipeline “provides a clear and actionable paradigm for automating the search for novel phenomena in large datasets.”
>
> Below, we respond to the reviewer’s specific comments and questions:
>
> ***
>
> > **The empirical validation of the method is limited. Despite showing success on 3 tasks (exclude toy data and CIFAR-10), the applications are drawn exclusively from the domains of physics and astrophysics, which is insufficient to substantiate the paper's broad claims of general scientific utility. Demonstrating efficacy in disparate fields, such as genomics, chemistry, or climate science, would be essential for this.**
>
> We agree that demonstrating our pipeline’s efficacy across disparate fields is important! A small point of clarification: we included baselines from three distinct scientific disciplines: particle physics (Figs. 2(a), 3(c), 7(b), 8(b)), astrophysics (LIGO, Figs. 2(b), 3(d), 5, 7(c), 8(c)), and histology (Figs. 3(e), 6, 7(d), 8(d)). Histology is distinct from physics and astrophysics.
>
> More importantly, these datasets span diverse **data modalities**: time-series (LIGO), point clouds (particle physics), and images (histology, CIFAR). This is reflected in the diversity of architectures used for the contrastive embeddings (transformers, resnets). We argue that, for our pipeline, this distinction is equally as important as conceptual distinctions between scientific domains. AutoSciDACT interacts with a discipline at the level of datasets, meaning a variation in modality is arguably the relevant distinction between scientific domains. Higher-level distinctions between domains are relevant insofar as they impact how well data from a particular domain can be embedded into a low-dimensional contrastive space, which is related to intrinsic qualities of data (e.g. information content, diversity of classes, availability of relevant augmentations, etc.) With this in mind, our experiments across modalities are broader in scope than the qualitative distinction between e.g. particle physics and astrophysics.
>
> While we argue that the presented datasets already span a broad range of scientific applications, we have conducted an additional study using a genomics dataset comprised of images of butterfly hybrids paired with genetic information. We trained AutoSciDACT using pictures of non-hybrid butterflies with labels identifying the butterflies as belonging to one of 12 subspecies classes. Butterflies form new hybrids in the wild, and we used our pipeline to detect "anomalies" (i.e. new hybrids) in the testing dataset. **This study will appear in the updated camera-ready version of our paper, though unfortunately its results (i.e. plots) cannot be shared here.**
>
> ***
>
> > **The paper's conceptualization of "scientific discovery" is limited, which narrows the framework's scope and makes its title potentially misleading. The term "discovery" often implies not just the detection of a new phenomenon, but also the generation of an explanatory hypothesis for it. The AutoSciDACT framework is tailored for the first step—identifying observational novelty—but its scope does not extend to the crucial subsequent step of building models or inferring the causal mechanisms that would explain the detected anomaly.**
>
> As correctly pointed out by the reviewer, this work focuses on the statistical strategy to detect significant anomalous behavior. It does not provide a solution for the interpretation and signal hypothesis generation that follow the detection stage. However, the NPLM algorithm outputs not only a scalar test but also a model shaping the log-density-ratio between the reference sample and the sample of interest. This information can be used to get insights on the shape and location of the anomaly: similarly to standard classifier-based tests, the model $f_{\hat{w}}$ resulting from the NPLM training can be sigmoided to obtain a classification score that can be used to select the most anomalous data points in the dataset; the reference set reweighted by the exponent of the model $f_{\hat{w}}$ represents the learnt density distribution of the data, giving information on the shape of the generative process. While these insights do not provide a unique theoretical interpretation for the signal, they can guide experts in hypothesis formulation.
>
> To demonstrate how this can be used in a real-data example, we have added a new appendix to the draft that uses particle collider data from the Higgs boson discovery (Open Data from the CMS experiment). In this study, we show that the most significant NPLM scores can point to the most anomalous events and indicate a Higgs boson. In particular, we find that the mass of the Higgs boson can be hinted at by examining correlations between the NPLM score and the four-lepton invariant mass among events flagged as most anomalous. Crucially, the learned embedding did not use four-lepton invariant mass as an input feature, but was still capable of flagging a mass overdensity via NPLM. It was trained instead on the raw inputs of the four leptons in an event (20 inputs in total). **This study will appear in the updated camera-ready version of our paper, though unfortunately its results (i.e. plots) cannot be shared here.**
>
> ***
>
> > **In Lines 201-202, the description of the synthetic dataset mentions rotating 'The full M × N dimension space.' This phrasing was slightly confusing, as my understanding is that N represents the number of distinct data clusters, while the data itself exists in a (D+M)-dimensional space. Could you please clarify if the text was intended to state that the full (D+M)-dimensional space is rotated to obscure the signal?**
>
> Thanks for catching this! You're right: the text should refer to the full D+M-dimensional space being rotated to obscure the signal. We corrected the typo in the manuscript.
>
> ***
>
> > **In Figure 2 (right panel), in the 2D projection, the anomalous signal class (pink) appears to have a significant visual overlap with one of the background classes (blue). This raises a question about the pipeline's effectiveness in this specific case. Could you elaborate on how detection is achieved here? For example, is the separation clearer in other, un-plotted dimensions of the embedding, or is the NPLM test sensitive enough to detect a subtle overdensity even within this overlapping region?**
>
> This is a great observation, and highlights one of the most important aspects of our pipeline. Indeed, in this example, the signal strongly overlaps with the background, imposing severe challenges for standard anomaly detection via e.g. flagging outliers. NPLM (our method), is explicitly designed as a two-sample test, which compares the dataset containing anomalous points to a reference dataset that is anomaly-free by construction at a distributional level. The test is sensitive to collective (distribution-level) anomalies that can manifest in many ways: as shape deviations/distortions/overdensities *in the bulk of the background-populated region*, or as excesses in the tails. In this case, we are sensitive to a distortion in the bulk. Since our method is sensitive to even these very subtle anomalies, it is a powerful general-purpose tool for discovering novelty in scientific datasets.
>
> ***
>
> > **In Figure 3 (d) and (f), in these two panels, the 'supervised' baseline (blue line) consistently outperforms the 'ideal-supervised' baseline (red line), which is the opposite of the trend seen in the other datasets. It's notable that both of these are image-based tasks. Could you offer some insight into this counter-intuitive result?**
>
> You're right that the supervised baseline outperforms the ideal supervised baseline in some cases, which is indeed counterintuitive. As a small point of clarification: panel (f) refers to the CIFAR dataset, which is indeed an image-based dataset, while panel (d) shows the results of the gravitational-wave dataset, which consists of time series (see Appendix A2, Figure 5).
>
> The difference between the two baselines lies in the contrastive pretraining; for the supervised baseline, the anomalous signal information is omitted, while in the ideal supervised setting, the anomalous signal is added as an additional class. Rather than the data modality, the number of classes drives the difference: if the number of classes is already high relative to the embedding dimension (as it is for the CIFAR-10 and gravitational wave datasets), embedding an additional class in the low-dimensional embedding ($d=4$) is more challenging and might be suboptimal, leading to the observed difference.
>
> We have also performed some additional studies exploring our pipeline’s performance with larger embedding dimensions $d = 8, 16, 32$, with all other factors (training time, NPLM kernel size, etc) kept the same as in the main results (Fig 3). In these studies we find that the "ideal supervised" performance quickly equalizes with or exceeds that of "supervised" as $d$ is increased. **This study will appear in the updated camera-ready version of our paper, though unfortunately its results (i.e. plots) cannot be shared here.**
>
> ***
>
> **Summary**
>
> We thank the reviewer again for their thoughtful, constructive feedback and insightful comments on our paper. We hope our response has clarified key details of our work and sufficiently addressed the reviewer's questions and concerns. Please let us know if we can further clarify, explain, or expand upon any specific points - we would be quite happy to continue a dialog during the discussion period!

---

> > ### Comment · Reviewer_jLRS · 2025-08-01
> >
> > Thank you for your detailed response. I think the concerns are adressed. I raised my rating to 4. Good luck with your submission!

---

### Official Review · Reviewer_drbe · 2025-07-03

**Clarity:** 3
**Significance:** 4
**Originality:** 3
**Rating:** 5
**Confidence:** 4

**Summary:**

The authors introduce AutoSciDACT (Automated Scientific Discovery with Anomalous Contrastive Testing), a new
two-part tool to detect and statistically validate anomalies in large and complex scientific datasets. The tool helps
solve two critical challenges in scientific discovery: the high dimensionality and noise typical in real-world data, and
the need for rigorous, interpretable statistical testing of potential outliers.
During pre-training, AutoSciDACT utilizes supervised contrastive learning to compress raw data into
low-dimensional, semantically meaningful embeddings. This will incorporate some domain knowledge by using
labeled data or scientifically relevant augmentations, which provide the necessary complexity reduction while
keeping the salient features of the raw data.
In the discovery phase, the pipeline applies the New Physics Learning Machine (NPLM), which is a kernel-based,
signal-agnostic statistical hypothesis testing methodology. NPLM allows us to compare an "observed" dataset to a
"reference" background dataset in order to identify any distributional shifts (e.g., clusters or overdensities) that may
represent new phenomena. The method outputs statistically calibrated significance scores (Z-scores, p-values),
supporting robust claims of discovery.
The authors evaluated AutoSciDACT against five datasets well within astronomy, particle physics, biology, and
computer vision , demonstrating its domain-agnostic performance and sensitivity to subtle signals—even at sub-1%
anomaly rates. The authors have also discussed its potential towards detection of dataset shift (e.g., CIFAR-10 vs.
CIFAR-10.1), suggesting broader utility in model evaluation and data quality assessment.

**Questions:**

1. How robust is AutoSciDACT to label noise or partial supervision in the pre-training phase?
Since the pipeline relies on supervised contrastive learning using labeled data, it's important to understand how
performance degrades with noisy or incomplete labels, especially in domains where precise annotation is difficult.
Suggestion: Consider including experiments (even synthetic) that measure sensitivity to label corruption or using
semi-supervised SupCon variants.
Impact: Demonstrating robustness (or proposing mitigation strategies) would increase confidence in the method’s
applicability to less curated domains.

2. Can the method handle domain shift between simulated/reference data and observed data?
The pipeline assumes the background data (R) and the test data (D) are drawn from the same distribution under the
null hypothesis. However, simulation-to-real gaps or data collection shifts are common in real scientific settings.
Suggestion: Provide a discussion or ablation showing how mild domain shift affects NPLM outputs or embeddings,
and whether techniques like domain adaptation or uncertainty modeling can be integrated.
Impact: If addressed well, this would significantly strengthen the generalizability and practical utility of
AutoSciDACT.

3. How sensitive is NPLM’s performance to the fixed 4D embedding choice?
While low-dimensional embeddings are necessary for statistical tractability, this constraint may lead to loss of
information, especially for datasets with many complex classes (e.g., CIFAR-10).
Suggestion: Include a sensitivity study on the embedding dimension (e.g., comparing d = 4, 8, 16) and how it affects
both detection performance and computational cost.
Impact: Clarifying this trade-off can help users balance interpretability vs. expressiveness and would inform how to
scale AutoSciDACT to larger problems.

**Ethical Concerns:**

["NO or VERY MINOR ethics concerns only"]

**Final Justification:**

The authors appropriately acknowledge and clarifies the issues I raised. My score stands.

**Limitations:**

Yes

**Paper Formatting Concerns:**

No major formatting issues found.

**Quality:**

4

**Strengths And Weaknesses:**

**Strengths**

Quality

* The paper is technically sound, with well-justified methodological choices. It combines contrastive learning
and New Physics Learning Machine (NPLM) into a coherent, end-to-end pipeline.
* Experimental validation is comprehensive across diverse domains (astronomy, physics, histology, synthetic
data, and images), demonstrating both robustness and generalizability.
* Statistical claims are backed by empirical and asymptotic p-values and Z-scores, following good scientific
practice.
* The authors are transparent about their assumptions, limitations (e.g., embedding dimension, domain
shifts), and the scope of the method.

Clarity

* The paper is well-organized and readable, particularly for readers familiar with ML.
* Diagrams, dataset descriptions, and appendix details contribute to a clear presentation of the pipeline and
experiments.
* The method's purpose (scientific novelty detection with statistical rigor) is clearly distinguished from
general anomaly detection approaches.

Significance

* This work addresses a critical and underexplored need in machine learning for science: tools that not only
detect anomalies but quantify their statistical significance in a scientifically interpretable way.

* AutoSciDACT offers a domain-agnostic framework applicable to any field with high-dimensional, labeled
data—making it valuable for physicists, biologists, astronomers, and potentially even ML safety
researchers.

Originality

* While the core components (contrastive learning, NPLM) are based on existing work, their integration into
a unified pipeline tailored for scientific discovery is novel and well-motivated.

* The use of Supervised Contrastive Learning for dimensionality reduction that respects domain labels is an
effective and underused idea.

* The pipeline is also shown to be extensible: e.g., in Section C, it is repurposed for evaluating dataset shift
and generative model fidelity, suggesting a broader set of applications beyond discovery.

**Weaknesses**

Quality

* The performance of AutoSciDACT relies heavily on the availability and quality of labeled background
data. In real-world domains with scarce or noisy labels, this assumption may limit applicability.

* Although the authors mention potential solutions (e.g., augmentations, domain knowledge), there is no
quantitative study on robustness to label noise or partial supervision.

* The embedding dimension is fixed to 4 for NPLM compatibility. While this keeps the statistical test
tractable, it may limit expressiveness, especially in highly multimodal data like CIFAR-10.

Significance

* Despite strong empirical results, AutoSciDACT currently assumes that the reference and observed
distributions are aligned. In real-world applications, domain shift between simulated (or historical)
reference data and observed data is common, and this limitation is only addressed as future work.

* The application to scientific discovery is exciting, but no real-world discovery example is included—only
simulated signals in known datasets. Including or proposing such a use case would strengthen the impact.

Originality

The core components (e.g., SupCon, NPLM) are re-used from prior work, and while the combination is
novel and useful, the method itself is not radically innovative in terms of algorithmic design.

---

> ### Author Rebuttal · Authors · 2025-07-31
>
> We sincerely thank the reviewer for their thoughtful and encouraging feedback. We are pleased to hear that our method was found to be “novel and well-motivated”. We especially appreciate your recognition that our pipeline not only detects anomalies but also quantifies their statistical significance in a scientifically interpretable way, as this was the central motivation of this work. Below, we respond to the specific weaknesses, suggestions, and questions you raised, and we clarify a few points to strengthen the paper further. We are confident that the revisions based on your feedback will help improve the manuscript further.
>
> ***
>
> > **The performance of AutoSciDACT relies heavily on the availability and quality of labeled background data. In real-world domains with scarce or noisy labels, this assumption may limit applicability.**
>
> > **How robust is AutoSciDACT to label noise or partial supervision in the pre-training phase?**
>
> These are great observations and questions! This is certainly a concern for scientific domains that rely on human- or algorithmically-labeled datasets where ground-truth is inaccessible. We agree that substantial label noise would degrade the quality of the learned embedding space, but NPLM’s sensitivity to subtle tails or overdensities may preserve its performance as long as a suitable background-only reference can be identified.
>
> To address this question, we have performed an additional study exploring the impact of label noise in the pre-training data using the JetClass dataset. We trained variants of the embedding with 1%, 2%, 5% and 10% label noise, where X% refers to the probability of a training sample’s label being randomly changed to a different value. During training, we observed that high label noise caused the models to learn poorly separated and less structured embeddings of the background classes. With a 1% injection of Higgs signal, we observed that the empirical NPLM significance across 500 toys degraded by about $1\sigma$ from 0% to 10% label noise, from $3\sigma$ to $2\sigma$. (The $3\sigma$ at 0% noise is limited by the choice of 500 toys; the asymptotic significance at this injection was around 4$\sigma$ and decreased to $2\sigma$ by 10% label noise). The Mahalanobis metric was more stable, but significantly less sensitive ($<2\sigma$). **This study will appear in the updated camera-ready version of our paper, though unfortunately its results (i.e. plots) cannot be shared here.**
>
> ***
>
> > **How sensitive is NPLM’s performance to the fixed 4D embedding choice?**
>
> This is a good question, and highlights some of the key design choices that went into developing AutoSciDACT. We agree that, by conventional representation learning standards, a four-dimensional embedding is relatively small. As you point out, this choice is made to keep the statistical analysis tractable. Exploring higher (but still modest) dimension embeddings is a great idea and will strengthen the claims made in our paper.
>
> To this end, we have performed additional studies exploring our pipeline’s performance with larger embedding dimensions $d = 8, 16, 32$, with all other factors (training time, NPLM kernel size, etc) kept the same as in the main results (Fig 3). At modest signal injections – where NPLM and any of the other baselines are able to detect an anomaly with $\sim2-3\sigma$ significance or higher – we observe fairly mild variation in NPLM’s performance up to $d = 32$. The sensitivity degrades slightly as dimension is increased, but by a nearly negligible amount (0 to 0.5$\sigma$) and not in all cases. The Mahalanobis distance, on the other hand, tends to perform better with increasing dimensionality. We hypothesize that this is due to more dimensions helping the background clusters be pushed further apart under the SimCLR objective. Since Mahalanobis is defined relative to these cluster centroids, a collection of tighter/better-separated clusters would be more sensitive to outliers. **This study will appear in the updated camera-ready version of our paper, though unfortunately its results (i.e. plots) cannot be shared here.**
>
> ***
> > **In real-world applications, domain shift between simulated (or historical) reference data and observed data is common, and this limitation is only addressed as future work.**
>
> > **Can the method handle domain shift between simulated/reference data and observed data?**
>
> This is an important observation - we fully agree with the reviewer on the importance of managing domain shifts when applying the methodology to real data. Our goal is to make this work directly applicable to real-world datasets. In light of that, we have chosen a variety of datasets as our benchmarks, some of which include real data (histology, gravitational-wave astronomy), while others are built on simulation.
>
> The histology datasets are “real” datasets, labelled by humans according to which organ they belong to and whether the fat-liver disease was present. While there may be elements of domain shift in the dataset due to variations in the data acquisition, such as staining, these elements are corrected through human labels. The choice of supervised contrastive learning then effectively mitigates the domain shift by explicitly using these labels. In the gravitational wave study, backgrounds are taken from signal-free regions of real data, and thus implicitly feature many of the realistic variations that might be present in another sample of real data analyzed using our pipeline. The signals are simulated and injected into the backgrounds, and reflect state-of-the-art modeling that is used in mainstream gravitational wave analyses. While domain shift is not *explicitly* addressed in either of these cases, we want to highlight that training dataset curation can indirectly have an impact. Significant domain adaptation strategies may not be necessary to perform anomaly detection searches if the training and reference datasets can be (partially) taken from real data,
>
> Lastly, we should mention that there are variants of NPLM that can handle systematic uncertainties (see e.g. Ref. [51] in our paper), and these uncertainties can help account for the impact of domain shift. Including this variant of NPLM is beyond the scope of this work, but we hope to continue building on our ideas in AutoSciDACT to extend it towards domain adaptation.
>
> ***
>
> **Summary**
>
> We thank the reviewer again for their thoughtful, positive, and constructive feedback. We hope that our responses have clarified key details of our work and addressed the reviewer's concerns. Please let us know if we can further clarify, explain, or expand upon any specific points - we would be quite happy to continue a dialog during the discussion period!

---

> > ### Comment · Reviewer_drbe · 2025-08-05
> >
> > The authors appropriately acknowledge and clarifies the issues I raised. My score stands.

---

### Official Review · Reviewer_2DCU · 2025-07-14

**Clarity:** 1
**Significance:** 2
**Originality:** 1
**Rating:** 2
**Confidence:** 4

**Summary:**

This article presents an anomaly detection approach that combines contrastive learning with hypothesis testing. The authors propose a general-purpose, two-stage pipeline for detecting novelty in scientific data, aimed at enabling end-to-end automatic scientific discovery. The pipeline is evaluated on datasets from diverse domains, including astronomy, particle physics, histology, and CIFAR datasets.

**Questions:**

* To clarify the technical details: for the histology data and CIFAR images, is the embedding dimension fixed to 4? If so, I’m concerned that this setting may not be optimal and could negatively impact the experimental results.

**Ethical Concerns:**

["NO or VERY MINOR ethics concerns only"]

**Final Justification:**

The paper lacks rigorous technical justification and thorough model evaluation. The proposed "baselines" consist of overly simplistic anomaly scores that do not engage with the contrastive training framework introduced in the first stage. Due to these methodological shortcomings, I cannot justify increasing my score.

**Limitations:**

See "Strengths And Weaknesses".

**Quality:**

2

**Strengths And Weaknesses:**

The paper introduces a two-stage pipeline combining contrastive learning and hypothesis testing for anomaly detection in scientific data. While the approach is promising, it falls short of demonstrating true generalist capabilities for automated scientific discovery. The methodology and experimental setup raise concerns about generalization, applicability, and evaluation rigor. Several technical choices and omissions limit the strength of the conclusions.

* The method is presented as a generalist approach for Automated Scientific Discovery, but in practice, it appears to be a specialized anomaly detection pipeline targeting a narrow class of anomalies.
* The paper lacks a comprehensive set of baseline comparisons. Several strong and commonly used anomaly detection methods are omitted, making it difficult to assess the relative performance of the proposed approach.
* Some settings and claims (e.g., universality across domains) are not well-motivated or empirically validated.
* The related work section is incomplete and does not adequately position the proposed method within the broader landscape of existing anomaly detection research.
* The choice of a simple MLP as the supervised baseline is insufficient for meaningful comparison, especially when stronger models could offer a better performance benchmark.

---

> ### Author Rebuttal · Authors · 2025-07-31
>
> We thank the reviewer for their feedback. We are concerned that some of the reviewer's critiques inaccurately reflect the content of our paper or miss crucial details. Some others make broad claims but without detailed justification or explanation, making it difficult to craft a detailed response. We will attempt to clarify some misunderstandings, and do our best to respond point-by-point to the reviewer’s comments.
>
> ***
>
> > **Pipeline target[s] a narrow class of anomalies” and “some settings and claims (e.g., universality across domains) are not well-motivated or empirically validated.**
>
> We demonstrate our pipeline on data from five distinct domains (one synthetic, one image-based, and three scientific; elaborated in section 4 and appendix A). Data modalities vary widely across these domains, and the anomaly takes a different form in each dataset. In section 6, we present results for all datasets (including time series data, images, point clouds, and tabular data), showing sensitivity to very small fractions of anomalous signals in each domain. We disagree with the reviewer’s characterization of these domains as a “narrow class of anomalies”, and point to our extensive experiments in each domain as empirical validation of our approach’s wide applicability.
>
> ***
>
> > **The paper lacks a comprehensive set of baseline comparisons. Several strong and commonly used anomaly detection methods are omitted[...]**
>
> As discussed in our Related Work section (Sec. 2), our anomaly detection approach differs from most standard approaches in its statistical rigor, with a p-value/Z-score being the primary figure of merit. To our knowledge, this approach has not been extensively explored in prior work. Standard approaches typically report e.g. AUROC and make no claims of statistical significance, making direct comparisons difficult. We include three baselines where a statistically rigorous anomaly score is possible: two “fully supervised” methods (MLP classifiers trained to find the anomaly) and the Mahalanobis distance.
>
> Additionally, we have performed studies with two additional baselines: Maximum Mean Discrepancy (MMD) and Frechet Distance (FD). We have evaluated these baselines for the gravitational wave, particle physics, and CIFAR datasets. We use the “empirical” method to compute p-values/z-scores with these metrics, as their asymptotic distributions are unknown. The results vary somewhat between datasets, but in all cases the empirical MMD/FD performs as well as or below empirical NPLM (our method) with the default $d = 4$ contrastive embedding space. MMD/FID and NPLM perform more equally at lower anomaly injection fractions, while NPLM becomes significantly better for larger injections. Interestingly, we do not observe consistent trends in the relative performance among the three non-NPLM methods (Mahalanobis, MMD, FID), other than that they tend to lag behind NPLM. This highlights the complementarity of the various statistical anomaly detection methods we compare with NPLM. **This study will appear in the updated camera-ready version of our paper, though unfortunately its results (i.e. plots) cannot be shared here.**
>
>
> ***
>
> > **The choice of a simple MLP as the supervised baseline is insufficient for meaningful comparison[...].**
>
> We agree that an MLP classifier would be insufficient if trained on the raw data inputs, but we argue this is not the relevant setting for discussing baselines for our approach. The foundation of our pipeline is the contrastive embedding, which vastly reduces the dimensionality of a dataset into a handful of expressive features. The anomaly detection step proceeds using these features rather than the raw inputs, which is crucial for allowing us to make statistically robust claims about the presence of anomalies (making statistically rigorous claims from high-dimensional raw inputs would require a prohibitively large number of samples).
>
> Given that dimensionality reduction is critical for the statistical methods used, we argue that it makes the most sense to compare anomaly detection methods applied to the embeddings rather than raw inputs. With this in mind, using an MLP as a supervised baseline is sensible: the data are vectors in a low-dimensional space, so the classification problem doesn’t require a more sophisticated architecture.
>
> ***
>
> > **The related work section is incomplete and does not adequately position the proposed method within the broader landscape of existing anomaly detection research.**
>
> We find the statement too vague to address meaningfully. No specific works, subfields, or methodological gaps are cited, making it difficult for us to act on this critique. Our related work section (section 2) was written with careful attention to representative categories in anomaly detection relevant for our work, including contrastive learning, contrastive anomaly detection, and hypothesis testing for anomaly detection, and we explicitly cite and discuss key works from each category. While most of the current literature focuses on identifying anomalous data, our work focuses on giving a meaningful statistical interpretation of said anomalies in a scientific context.
>
> ***
>
> > **To clarify the technical details: for the histology data and CIFAR images, is the embedding dimension fixed to 4? If so, I’m concerned that this setting may not be optimal and could negatively impact the experimental results.**
>
> The embedding dimension is fixed to 4 for experiments presented in the main text, as noted in Section 5. We have performed additional studies increasing the dimensionality of the embedding up to 32. We find no significant change in detection performance. In realistic applications where the signal is not known a priori, tuning the dimensionality of the embedding is not possible, and a priori design choices must be made. As discussed in the paper, a low-dimensional embedding was selected to make the statistical component of the anomaly detection tractable. In addition, low-dimensional representations allow the deployment of classical statistical methods and benefit uncertainty quantification.
>
> Our additional studies with dimensions $d = 8, 16, 32$ held all other factors (training time, NPLM kernel size, etc) the same as in the main results (Fig 3). At modest signal injections – where NPLM and any of the other baselines are able to detect an anomaly with $\sim2-3\sigma$ significance or higher – we observe fairly mild variation in NPLM’s performance up to $d = 32$. The sensitivity degrades slightly as dimension is increased, but by a nearly negligible amount (0 to 0.5$\sigma$) and not in all cases. The MMD and FD metrics are similarly stable as a function of dimension, varying only slightly and sometimes non-monotonically up to $d=32$. The Mahalanobis distance, on the other hand, tends to perform slightly better with increasing dimensionality. We hypothesize that this is due to more dimensions helping the background clusters be pushed further apart under the SimCLR objective. Since Mahalanobis is defined relative to these cluster centroids, a collection of tighter/better-separated clusters would be more sensitive to outliers.
>
> ***
>
> **Summary**
> We thank the reviewer again for their feedback, and sincerely hope that our explanations and clarifications above have helped improve the paper and address the reviewer's concerns. We are happy to follow up with additional details during the discussion period!

---

> > ### Comment · Reviewer_2DCU · 2025-08-09
> >
> > I would like to thank the authors for their response. Statistical interpretation could be done by choosing an optimal anomaly score for the methods that were using AUCROC as the figure of merit. Including these as baselines will make the methodology more solid. And I am still skeptical about the embedding dimension of 4. The embedding dimension could at least be optimized using a validation set at the first contrastive learning stage. And I suppose the optimal dimension would vary for different types of data and encoder architectures, especially for CIFAR that is natural images with high input data dimensionality.

---

### Public Comment · ~Philip_Harris1 · 2026-01-23
**Update to incorrect citation**

The authors noted that a mistake was made in reference 12 in the paper. This reference was pointed out by

``Nazar Shmatko, Alex Adam, and Paul Esau. "GPTZero finds 100 new hallucinations in NeurIPS 2025 accepted papers", January 21, 2026``  https://gptzero.me/news/neurips/

We apologize for the mistake in references. The mistake came about from the fact that we utilized LLMs to help generate the associated BibTeX of the references, based on contextual cues, in this case, the lead author and the first word of the title. We have included an acknowledgment of this mistake in the paper, along with the corrected reference. We stand by the scientific integrity of this paper.

In the interest of full transparency, we are currently in the process of

  * Updating the arxiv with a second version listed here (https://www.arxiv.org/abs/2510.21935)
  * Contacting NeurIPS'25 program chiars
  *  We include the updated paper in a link here: https://tinyurl.com/3zazxudw

The details of the changes are listed below.  We sincerely apologize to the entire community and the authors of the incorrectly cited paper, and we thank the GPTZero team for bringing this issue to our attention.

The listed reference in the Related Works section gives a reference to:

```
@article{azabou2022mineclip,
  title={MINECLIP: Multimodal neural exploration of CLIP latents for automatic video annotation},
  author={Azabou, Mehdi and Weber, Micah and Ma, Wenlin and others},
  journal={arXiv preprint arXiv:2210.02870},
  year={2022}
}
```
Which includes an incorrect 2nd and 3rd author, an incorrect title, and an arXiv reference. The correct reference that we intended to cite

```
@article{Azabou2021MineYourOwnView,
  title={Mine your own view: Self-supervised learning through across-sample prediction},
  author={Azabou, Mehdi and Azar, Mohammad Gheshlaghi and Liu, Ran and Lin, Chi-Heng and Johnson, Erik C and Bhaskaran-Nair, Kiran and Dabagia, Max and Avila-Pires, Bernardo and Kitchell, Lindsey and Hengen, Keith B and others},
  journal={arXiv preprint arXiv:2102.10106},
  year={2021}
}
```

---

### Note · Authors · 2025-08-11

We thank the organizers for introducing the final remark option. We had positive and productive interactions with three of our four reviewers, who highlighted the novelty of our statistically rigorous approach to anomaly detection and its relevance in scientific applications. Their constructive feedback helped strengthen our work. Unfortunately, reviewer 2DCU did not provide detailed or actionable critique, and replied to our rebuttal just 5 minutes before the deadline. We are thankful for this additional opportunity to reply further below:

***

> **Statistical interpretation could be done by choosing an optimal anomaly score for the methods that were using AUCROC as the figure of merit.**

Choosing an “optimal” score is more akin to a *supervised* method - it would require prior knowledge about the anomaly, which is not the scenario addressed in this work. Alternatively, it would require splitting the data of interest into two subsets for separate training and testing. We argue that in the presence of low sample sizes and/or rare signal injections, splitting significantly undermines detection sensitivity (e.g. Ref. 50 cited in the manuscript, where NPLM far outperforms AUCROC). Our approach requires no tuning and is designed to detect *broad classes of anomalies that are unspecified a priori*.

***

> **I am still skeptical about the embedding dimension of 4. The embedding dimension could at least be optimized using a validation set [...] I suppose the optimal dimension would vary for different types of data and encoder architectures, especially for CIFAR [...]**

As noted above, data scarcity and signal rarity in multiple scientific contexts do not easily allow validation sets. As detailed in our rebuttal, we performed an additional study scanning the embedding dimension up to 32, finding very little variation in performance. A significant result of this work is the strong sensitivity we achieve with low-dimensional embeddings. Since there is no *a priori* “optimal” dimensionality for unknown anomalies, and keeping in mind the intended application to real scientific problems, low dimensionality ensures that downstream scientific analysis (e.g. modeling epistemic uncertainties) remains tractable. Furthermore, our method performs strongly on 4-dimensional CIFAR-10 embeddings, and in the appendix we show that it even detects domain shift between CIFAR-10 and CIFAR-5m.

---

### Decision · Program_Chairs · 2025-09-17

**Decision:**

Accept (poster)

**Comment:**

The work proposes a method for detecting anomalous sample set of scientific artifacts given an established dataset of normal artifacts of the same kind. The method combines dimensionalitybreduction via contrastive learning and a recently published statistical hypothesis testing technique called NPLM. Most reviewers note that the paper is well written and contains an extensive empirical evaluation.

However, despite voting for acceptance, the reviewers point out that the advertises scientific discovery *in datasets* is not addressed in the paper, and the paper explorers comparison *berween* dataset. The technique appears to be quite standard on the high level: dimensionality reduction and hypothesis testing in low dimensional space. Actual scientific discovery has not been demonstrated in the case studies, and was instead simulated by injection of perturbed instances or instances of a missing class.

My known reading also raises certain concerns on the soundness of the statistical analysis. Z score in four dimensions does not have the same meaning  as in one dimension. Even under isotropic normal distribution, the confidence intervals are different.